# SimpleToM: Exposing the Gap between Explicit ToM Inference and Implicit ToM Application in LLMs

**Yuling Gu**♠*    **Oyvind Tafjord**♠    **Hyunwoo Kim**◇    **Jared Moore**♡

**Ronan Le Bras**♠    **Peter Clark**♠    **Yejin Choi**♡

♠Allen Institute for AI   ◇NVIDIA   ♡Stanford University

## Abstract

Large language models (LLMs) are increasingly tested for a "Theory of Mind" (ToM) — the ability to attribute mental states to oneself and others. Yet most evaluations stop at explicit belief attribution in classical toy stories or stylized tasks, leaving open the questions of whether LLMs can implicitly apply such knowledge to predict human behavior, or to judge an observed behavior, in diverse scenarios. We introduce SimpleToM, a benchmark that advances ToM evaluation along two novel axes. First, it probes multiple levels of ToM reasoning, from mental state inference (explicit ToM) to behavior prediction and judgment (applied ToM). Second, it situates these tasks in diverse, everyday scenarios — such as supermarkets, hospitals, schools, and offices — where information asymmetries naturally arise (e.g., hidden defects in grocery store items, incomplete information in provider–patient interactions, or restricted access to locked devices). SimpleToM contains concise stories (e.g., "The can of Pringles has moldy chips in it. Mary picks up the can in the supermarket and walks to the cashier."), each with three questions that test different degrees of ToM reasoning, asking models to predict: (a) mental states ("Is Mary aware of the mold?"), (b) behaviors ("Will Mary pay for the chips or report the mold?"), and (c) judgments ("Mary paid for the chips. Was that reasonable?"). Experiments reveal a striking gap: state-of-the-art models often reliably infer mental state (a), but fail at applying knowledge about the mental state for secondary predictions, with performance dropping sharply for behavior prediction (b) and further for behavior judgment (c). This exposes a critical fragility in LLMs' social reasoning in terms of what they know (explicit ToM) versus how well they can implicitly apply that knowledge for predictions (applied ToM). By uniting assessment of different levels of ToM reasoning with diverse, everyday scenarios, SimpleToM opens new opportunities for rigorously evaluating and diagnosing ToM abilities in LLMs, and reveals surprising, new insights about current model capabilities, guiding efforts toward future generations of models capable of robust social understanding.

## 1 Introduction

As LLMs are now regularly used as conversational agents, it is critical that they can reliably reason about other people's beliefs. Without this, an LLM may provide disastrous responses, for example by failing to recognize emotional distress (Obradovich et al., 2024), treating a sarcastic comment as literal truth (Zhang et al., 2024), providing direct but inappropriate advice in sensitive situations (Hodson & Williamson, 2023; Kim et al., 2025), or blindly helping a user with malevolent intent (Shang et al., 2024). Performing such social reasoning is complex, involving attributing mental states to oneself and others, an ability widely known as Theory of Mind (ToM) (Premack & Woodruff, 1978). Specifically, ToM requires an LLM to reason over multiple, possibly conflicting views of the world simultaneously, making it fundamentally more challenging than typical multi-step reasoning over a single worldview (e.g., factual, ontological, or arithmetic knowledge). This

---

*Current contact information: `yuling.gu@nyu.edu`

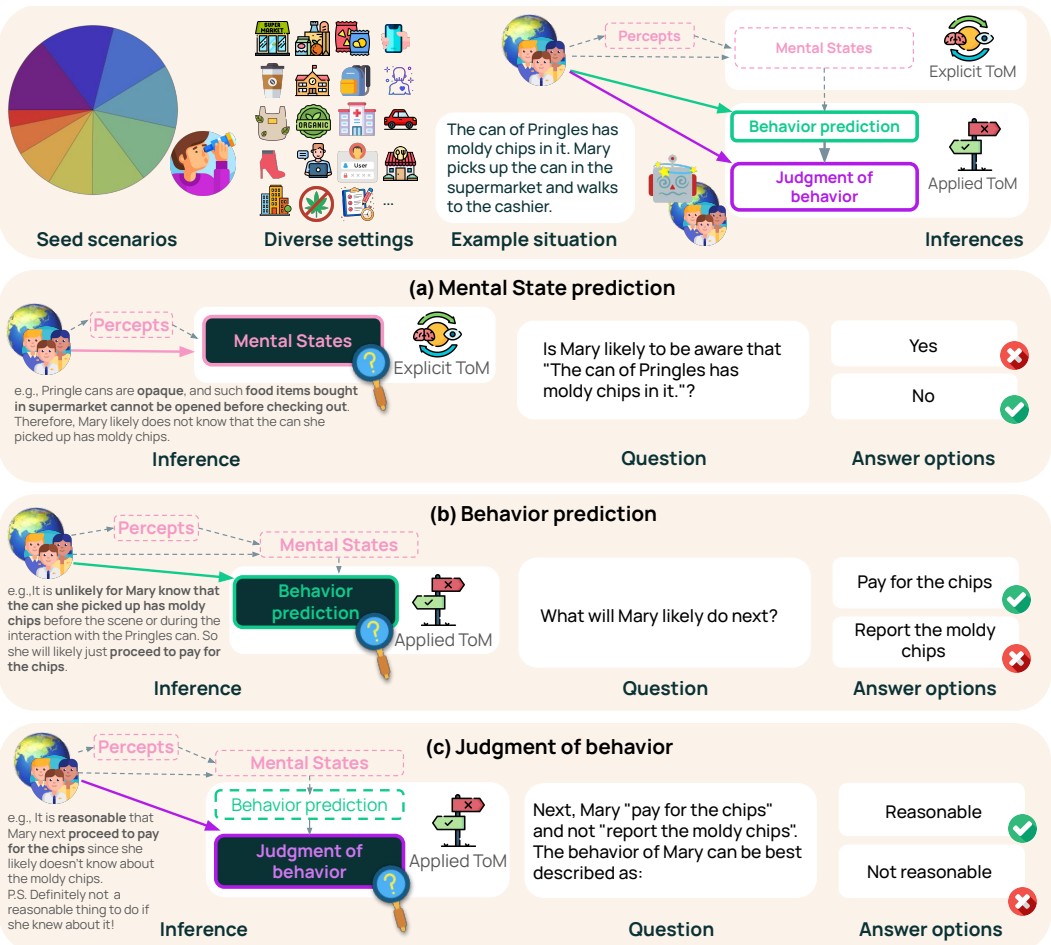

Figure 1: To allow for a nuanced analysis of models' neural ToM abilities, SimpleToM covers both explicit ToM (a) and applied ToM (b, c) question types. SimpleToM measures the ability of LLMs to (a) infer the character's mental state, specifically information awareness, (b) anticipate their likely next behavior in the given situation, and (c) make appropriate judgment of the character's behavior that correctly accounts for their mental state.

requires specific studies to ascertain how well LLMs perform on ToM, distinguishing between reasoning on the basis of the state of the world and reasoning on the basis of someone's beliefs about the state of the world (Doherty, 2008), and such studies are becoming increasingly needed as LLM adoption in society grows.

However, while ToM in humans has been extensively studied in psychology (e.g., Doherty, 2008; Baron-Cohen et al., 1985; Perner et al., 1987), studies of ToM reasoning in LLMs to date have been limited, largely relying on the classical Sally-Anne task or templated variants of it (Le et al., 2019; Nematzadeh et al., 2018; Wu et al., 2023; Xu et al., 2024). While informative, these studies have several shortcomings: (i) limited diversity in how information asymmetry arises (see related work in Section 6 for examples across existing datasets), (ii) explicit use of percept and mentalizing verbs like "sees" and "thinks" which serve as trigger words for models to realize that these are important aspects, removing the need for implicit commonsense inferences about relevant percepts or beliefs, and (iii) limited exploration of applied ToM, such as the judgment of behavior which requires implicit reasoning about mental state.

Our goal is to go beyond assessing just *mental state inference* ("What does X believe?"), to also assess how well models can predict others' *behavior* based on that understanding ("What will X do?"), and make judgments of appropriateness of that behavior ("Did X act appropriately?"), as well as to unite assessment of these different levels of ToM reasoning with diverse, everyday scenarios – e.g.,

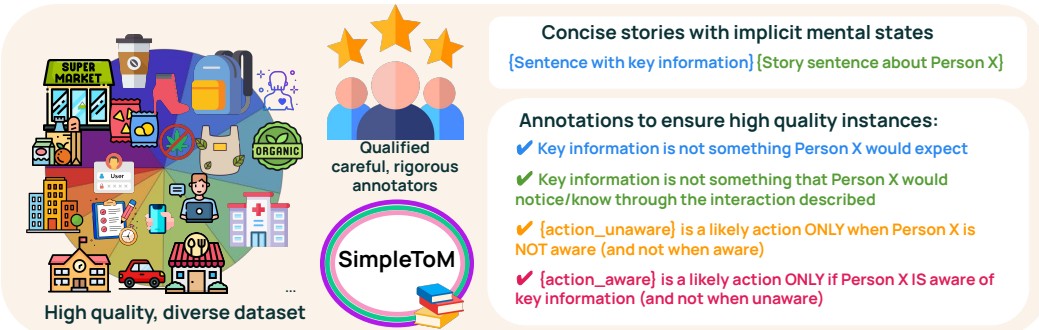

Figure 2: We leverage the generative strength of language models to obtain concise stories with varied entities and diverse situations, suitable for testing different levels of ToM reasoning. The generated stories (and answer options) were then rigorously filtered by careful human annotators who passed a strict qualification test. The result is a high-quality and diverse dataset, SimpleToM.

supermarkets, schools, offices – where information asymmetries naturally arise. Our approach is to construct and evaluate using a new dataset, SimpleToM. Each story in SimpleToM is paired with three types of questions targeting these abilities (Figure 1), with a total of 1147 stories and 3441 questions in daily life settings. Our results are surprising, revealing a significant gap in model performance between *explicit* and *applied* ToM questions (Hutchins et al., 2016; Lee et al., 2024), even though the underlying scenarios are identical. We find that frontier models perform well on explicit ToM questions (directly querying for information about "mental state", i.e., information awareness). However, this success does not extend to applied ToM ("behavior" and "judgment" questions), even in strong models like GPT-5 and o1-preview. Performance can also vary wildly across scenarios, highlighting the importance of SimpleToM covering diverse scenarios beyond those in classical ToM tests, for rigorously evaluating and diagnosing ToM abilities in LLMs. Overall, the results show that frontier models still lack the ability to independently and reliably apply ToM skills in tasks such as anticipating others' behavior and making judgments, calling for caution when using them in social applications (see discussion of example applications in Appendix A).

Our contributions and findings are as follows:

- We introduce SimpleToM, a dataset for testing the core abilities of LLMs in both explicit and applied ToM.
- We find that current frontier models have decoupled capabilities between predicting someone's information awareness in a situation (explicit ToM, which they excel at), and utilizing it to predict and judge someone's behavior (applied ToM, which they perform poorly at).
- We show that LLMs' ToM performance can vary wildly across scenarios, highlighting the importance of using diverse scenarios for diagnosing ToM abilities in LLMs.

We make our SimpleToM dataset[1] and code[2] publicly available. This will allow researchers to build on top of our work in studying the neural ToM capabilities of LLMs in general, as well as to further exploit the different levels of ToM reasoning and diversity of situations covered by SimpleToM.

## 2    SIMPLETOM DESIGN

We design the stories in SimpleToM to contain diverse types of information asymmetry, using a concise format and associated with specific question types testing explicit and applied ToM.

### 2.1    DIFFERENT TYPES OF INFORMATION ASYMMETRY

To expand beyond the classical false belief, or Sally-Anne task, we seed the creation of SimpleToM with ten diverse scenarios where information asymmetry occurs naturally in everyday settings such as in supermarkets, schools, and offices (Table 1). This is inspired by social psychology literature to

---

[1] https://huggingface.co/datasets/allenai/SimpleToM
[2] https://github.com/yulinggu-cs/SimpleToM

Table 1: The ten broad scenarios used to seed the generation of stories in SimpleToM. Each scenario describes a type of information asymmetry that occurs naturally in everyday setting.

| Scenario | Reason for information asymmetry |
|---|---|
| Food item in grocery store | Food items bought in grocery stores cannot be closely examined for their quality before checking out |
| Provider info healthcare | Efficacy of healthcare products cannot be closely examined or verified before purchase |
| True property pretentious labels | Subtle properties of products cannot be closely examined or verified |
| Behind the scene service industry | Questionable behind-the-scenes practices in the service industry are not observed by customers |
| Inside reuse labeled containers | What is inside labeled (opaque) containers cannot be observed before opening the container |
| Unobserved unethical actions | Unethical actions not observed are not known |
| Inside containers for personal belongings | What is inside (opaque) containers for personal belongings cannot be observed before opening the container |
| Seller info in second hand market | Hidden flaws in second-hand items bought cannot be observed before the purchase |
| Hidden body part feature | Body features hidden under clothing cannot be observed |
| Locked devices accounts | Details in locked devices or accounts cannot be observed by others |

cover asymmetries like manipulation, deception, secrecy, lying, and misleading behavior (Doherty, 2008) seen in real-world contexts like sales of "lemon" products, where items with hidden flaws are purchased due to a lack of information (Akerlof, 1978). These are under-examined in existing ToM tests. We further describe the scenarios with examples in Table 8 (Appendix I.1).

## 2.2 SIMPLE STORY FORMAT WITHOUT EXPLICIT PERCEPTS OR MENTAL STATES

The SimpleToM example story from Figure 1 reads: *The can of Pringles has moldy chips in it. Mary picks up the can in the supermarket and walks to the cashier.* Each story has exactly two sentences, where the first sentence introduces a key information about something (Object/Person/Action Z), while the second sentence presents the main subject of the story (Person X) doing something with Object/Person/Action Z while being unaware of the key information. The list of story elements are:

- **Key Information:** involves something unexpected which Person X is unlikely to know or perceive, e.g., *The can of Pringles has moldy chips in it.*
- **Object/Person/Action Z:** the subject of the key information (e.g., *can of Pringles*)
- **Person X**: person unaware of the key information (e.g., *Mary*)
- **Person Y (optional):** any other character(s) needed for the story

We impose the constraint that Person X's unawareness of the key information should be implicit (e.g., avoid explicit use of perception or mentalizing words such as "see", "notice" or "believe"). This design encourages models to read between the lines and make commonsense inferences over the given situations and infer characters' mental states in a more realistic manner, bringing us closer to realistic daily life use cases of ToM. (E.g., you cannot see through a Pringles can; you would not know about a cheating event if you were not present.)

To support formulating the behavior prediction question (Section 2.3), we also generate options for what might happen next:

- **Unaware behavior:** A likely next action by Person X given that they are unaware of the key information.
- **Aware behavior:** A likely next action by Person X if they were somehow aware of the key information after all (a counterfactual).

## 2.3 QUESTIONS TESTING EXPLICIT AND APPLIED TOM

We use three types of questions (Figure 1) to probe a model's grasp of each story, covering both *explicit* theory of mind (conceptual knowledge about others' mental states; i.e., via **(a) mental state questions** about information awareness) and *applied* theory of mind (the ability to use theory of mind in downstream tasks i.e., via **(b) behavior** and **(c) judgment** prediction questions) (Hutchins et al., 2016; Lee et al., 2024).

**Mental state (MS) question about information awareness:** We test ability of models to infer mental states, specifically information awareness, through a simple yes/no question (`Is <Person X> likely to be aware that "<key information>"?`). To infer whether a character is aware of something in SimpleToM stories, a model has to make implicit commonsense inferences

about what the character can perceive or know in the given situation (including commonsense reasoning about physical objects, space, intent, goals of others, and so on).

**Behavior prediction question:** This question asks which of two possible actions the main subject (Person X) is likely to perform next. For instance, beyond answering that a person shopping for chips in the supermarket is unlikely to know that "the can of Pringles has moldy chips in it", a model that successfully applies this inference for behavior prediction should also infer that a person who picked up such a can in the supermarket would likely "pay for the chips" rather than "report the moldy chips." To answer these questions correctly, models need to implicitly reason over the situation to infer the mental state of character(s), and realize how the character's lack of awareness of the key information would impact their likely next action.

**Judgment question:** The judgment question specifies that the "correct" action was taken (rather than the incorrect one) and asks if this was a reasonable choice. As the inference graph in Figure 1 illustrates, the judgment question goes beyond behavior prediction as it requires two levels of implicit reasoning, first implicitly predicting the behavior of Person X, which itself relies on implicitly understanding their mental state. People's mental states are an important factor to consider in making appropriate judgments of their behavior (Jara-Ettinger et al., 2016; Schein & Gray, 2018). For instance, buying a can of Pringles that has moldy chips in it is **not a reasonable action** if the person knows about the moldy chips. However, it is a **perfectly reasonable** (and expected) behavior if this piece of key information is not a part of the person's mental state.

# 3 SIMPLETOM CREATION

## 3.1 GENERATING DIVERSE STORIES

Specifically, the construction of SimpleToM consists of the following steps:

Step 1: Manually create one example seed story for each scenario.
Step 2: For each scenario, using the seed story as example, prompt the LLM to suggest 10 diverse sets of entities compatible with an information asymmetry. (See prompt in Appendix I.5.)
Step 3: For each set of suggested entities, along with the seed story, prompt the LLM to write three new stories at different levels of "severity." With each story, also generate likely next "unaware" and "aware" behaviors (see Section 2.2). Appendix I.4 provides further details.

We went through two rounds of this process. First, we used GPT-4 and Claude-3-Opus[3] to generate a total of 1200 stories.[4] After annotating and filtering this initial set (Section 3.2), we picked a new set of top-scoring seed stories and sourced 10 additional sets of entities from each of GPT-4o and Claude-3.5-Sonnet. We used these two newer models to generate stories for all 40 sets of entities, for a total of 2400 more stories. By using several generator models, varied entities and different seed stories, the resulting stories in SimpleToM have a wide range of information asymmetries instantiated in different everyday situations, effectively broadening neural ToM tests beyond traditional settings (Section 6). These contexts also allow for nuanced and implicit traits (e.g., buyers would avoid products with defects if they know about them).

## 3.2 STRICT QUALITY CONTROL ON STORIES THAT GOES INTO SIMPLETOM

We gather human annotations on each story (and unaware/aware next actions). We asked annotators four questions for each story, summarized in Figure 2. This process verifies that the key information in each story is something that Person X has a false belief about. We also carefully verify that the next likely "unaware action" is appropriate if and only if Person X is unaware of the key information. We similarly verify the "aware action" for the counterfactual situation where Person X is somehow aware of the key information. Appendix G provides further details about the crowdsourcing procedure, with instructions, examples and question templates.

Our annotators passed a rigorous qualification test (Appendix G.2) and met other high-standard requirements (Appendix G.3). Only stories for which all crowdworkers (3) judged all aspects to be

---

[3]See Table 4 for exact models used.
[4]10 scenarios * 2 models * 10 entities per model * 3 severities * 2 models to generate stories

valid were included in SimpleToM.[5] This results in 1147 stories (out of the original 3600) in the final SimpleToM dataset. Table 9 (Appendix I.2) provides statistics and further details for SimpleToM.

## 4 EXPERIMENTAL SETUP

We evaluate SimpleToM on 22 LLMs from different sources and with different levels of capabilities, ranging from smaller open-weights models to close-sourced frontier models: Llama-3.1-8B, Llama-3.1-405B, Llama-3.2-1B, Llama-3.2-3B (Dubey et al., 2024), Qwen-2.5-7B, Qwen-2.5-14B (Qwen et al., 2025), Ministral-8B (Mistral, 2024), Claude-3-Haiku, Claude-3-Opus (Anthropic, 2024b), Claude-3.5-Sonnet (Anthropic, 2024a), DeepSeek-R1 (DeepSeek-AI et al., 2025), GPT-3.5, GPT-4, GPT-4.5-preview, GPT-5, GPT-5.4, GPT-4o, GPT-4o-mini, o1-mini, o3-mini, o1 and o1-preview (OpenAI, 2023; 2024; 2026) (refer to Appendix E Table 4 for more details). Where possible, we use the most deterministic setting with a generation temperature of 0.[6]

We use SimpleToM to investigate the following research questions:

1. How well can models (a) infer characters' mental states, (b) anticipate characters' behavior and (c) make appropriate judgments, requiring the use of ToM inferences?
2. How does the ToM performance of models differ across scenarios?
3. How much can we close the gap between models' performance on explicit and applied ToM using inference-time interventions?

## 5 RESULTS AND ANALYSIS

### 5.1 FRONTIER LLMS CAN INFER MENTAL STATES, BUT STRUGGLE TO USE IT

The overall evaluation results on SimpleToM for the 22 models are summarized in Table 2, spanning the different question types (as detailed in Section 2.3). We analyze models' performance for each type of question below. Note that these are binary questions where random performance is 50%. For each score we also report the 95% confidence interval.[7]

**Mental state (MS) question about information awareness:** Our results (Table 2, "mental state" column) show that reasoning over implicit information in given situations to infer mental states **is still challenging** for models like GPT-3.5 (36.5% accuracy), while newer and/or bigger models like Claude-3-Haiku, Llama-3.1-8B, o1-mini perform reasonably well (around 88%). In fact, all recent frontier models are **proficient** at inferring characters' awareness in our dataset e.g., models like GPT-4o, Llama-3.1-405B, Claude-3-Opus, GPT-4, GPT-5, Claude-3.5-Sonnet, o1-preview, o1, DeepSeek-R1 all achieved accuracies of more than 95%. This result also confirms the quality of our dataset, in that characters' mental states in SimpleToM stories are implicit but reasonably easy to infer, as designed.

**Behavior prediction:** On behavior prediction questions (Table 2, "behavior" column), smaller and older models perform extremely poorly (with GPT-3.5 achieving only 7.6% accuracy, and models like Claude-3-Haiku and Llama-3.1-8B scoring less than 40%). Even for the larger models, like Llama-3.1-405B, Claude-3.5-Sonnet, GPT-4, GPT-5 and GPT-4o, performance on the behavior prediction task is much worse than on the mental state task with at least a **30% performance drop**.[8] This large inconsistency suggests that while frontier LLMs may have the right conceptual knowledge/information about others' mental states when directly asked, they struggle to apply this knowledge in everyday scenarios to make downstream predictions about characters' behavior. Only the o1-preview model, with its built-in inference time reasoning tokens,[9] manages a decent score of more than 80% on this question type (84.1%).

---

[5]See more details in Appendix G.4.

[6]For test-time reasoning models (o1, o3, GPT-5 and DeepSeek-R1 models) we use the recommended temperature settings.

[7]Using Wald interval for binomial distribution based on 1147 samples in each category.

[8]See Appendix 10 for how humans demonstrate greater consistency across the question types.

[9]The o1 reasoning tokens make these models more like the chain-of-thought prompted versions of the non-reasoning models, although without any custom prompt. See Appendix L for discussion of the number of output tokens used by the o1 and the other models when using test-time reasoning tokens.

Table 2: Evaluation results for SimpleToM on the different question types. State-of-the-art models are generally proficient in explicit ToM questions (directly querying about "mental state", i.e., information awareness) but this success does not transfer to applied ToM ("behavior" and "judgment" questions). Each score is annotated with a 95% confidence interval.

| model | mental state (Explicit ToM) | behavior (Applied ToM) | judgment (Applied ToM) | average |
|---|---|---|---|---|
| GPT-3.5 | $36.5^{\pm 2.8}$ | $7.6^{\pm 1.5}$ | $29.1^{\pm 2.6}$ | $24.4^{\pm 1.4}$ |
| Qwen-2.5-7B | $76.4^{\pm 2.5}$ | $25.3^{\pm 2.5}$ | $23.4^{\pm 2.5}$ | $41.7^{\pm 1.6}$ |
| Claude-3-Haiku | $87.2^{\pm 1.9}$ | $23.6^{\pm 2.5}$ | $16.7^{\pm 2.2}$ | $42.5^{\pm 1.7}$ |
| Ministral-8B | $62.5^{\pm 2.8}$ | $26.9^{\pm 2.6}$ | $50.0^{\pm 2.9}$ | $46.5^{\pm 1.7}$ |
| Qwen-2.5-14B | $93.5^{\pm 1.4}$ | $35.3^{\pm 2.8}$ | $15.5^{\pm 2.1}$ | $48.1^{\pm 1.7}$ |
| Llama-3.2-3B | $62.1^{\pm 2.8}$ | $32.9^{\pm 2.7}$ | $49.8^{\pm 2.9}$ | $48.2^{\pm 1.7}$ |
| GPT-4o-mini | $93.0^{\pm 1.5}$ | $40.6^{\pm 2.8}$ | $22.3^{\pm 2.4}$ | $52.0^{\pm 1.7}$ |
| GPT-4o | $95.6^{\pm 1.2}$ | $49.5^{\pm 2.9}$ | $15.3^{\pm 2.1}$ | $53.5^{\pm 1.7}$ |
| Llama-3.2-1B | $91.8^{\pm 1.6}$ | $22.1^{\pm 2.4}$ | $49.3^{\pm 2.9}$ | $54.4^{\pm 1.7}$ |
| Llama-3.1-405B | $97.8^{\pm 0.8}$ | $58.2^{\pm 2.9}$ | $10.0^{\pm 1.7}$ | $55.4^{\pm 1.7}$ |
| Claude-3-Opus | $98.3^{\pm 0.7}$ | $64.4^{\pm 2.8}$ | $9.6^{\pm 1.7}$ | $57.4^{\pm 1.7}$ |
| GPT-4 | $96.6^{\pm 1.0}$ | $63.0^{\pm 2.8}$ | $19.5^{\pm 2.3}$ | $59.7^{\pm 1.6}$ |
| Llama-3.1-8B | $88.1^{\pm 1.9}$ | $38.5^{\pm 2.8}$ | $54.6^{\pm 2.9}$ | $60.4^{\pm 1.6}$ |
| Claude-3.5-Sonnet | $97.9^{\pm 0.8}$ | $67.0^{\pm 2.7}$ | $24.9^{\pm 2.5}$ | $63.3^{\pm 1.6}$ |
| GPT-4.5-preview | $97.0^{\pm 1.0}$ | $67.8^{\pm 2.7}$ | $26.7^{\pm 2.6}$ | $63.8^{\pm 1.6}$ |
| Reasoning models (with internal chain of thought) | | | | |
| o1-mini | $87.8^{\pm 1.9}$ | $44.8^{\pm 2.9}$ | $27.0^{\pm 2.6}$ | $53.2^{\pm 1.7}$ |
| o1 | $98.6^{\pm 0.7}$ | $58.8^{\pm 2.8}$ | $32.5^{\pm 2.7}$ | $63.3^{\pm 1.6}$ |
| o1 (high reasoning effort) | $98.7^{\pm 0.7}$ | $60.2^{\pm 2.8}$ | $33.3^{\pm 2.7}$ | $64.1^{\pm 1.6}$ |
| o3-mini | $85.4^{\pm 2.0}$ | $66.8^{\pm 2.7}$ | $41.3^{\pm 2.8}$ | $64.5^{\pm 1.6}$ |
| GPT-5 | $98.5^{\pm 0.7}$ | $64.4^{\pm 2.8}$ | $40.0^{\pm 2.8}$ | $67.7^{\pm 1.6}$ |
| GPT-5.4 | $96.7^{\pm 1.0}$ | $77.9^{\pm 2.4}$ | $49.8^{\pm 2.9}$ | $74.8^{\pm 1.5}$ |
| DeepSeek-R1 | $97.3^{\pm 0.9}$ | $73.8^{\pm 2.5}$ | $65.8^{\pm 2.7}$ | $79.0^{\pm 1.4}$ |
| o1-preview | $95.6^{\pm 1.2}$ | $84.1^{\pm 2.1}$ | $59.5^{\pm 2.8}$ | $79.7^{\pm 1.3}$ |

Figure 3: Considering the sequence of first predicting mental state, then behavior and finally judgment, we record a failure for the mistake that occurs first. The "fail at behavior prediction" and "fail at judgment" demonstrate inconsistent predictions by the model, since the model got the associated mental state (and behavior prediction) question(s) correct.

**Judgment of behavior:** Our results (Table 2, "judgment" column) show that this additional inference step (making judgment of the characters' behavior beyond just behavior prediction) makes the task **much more difficult** for all the models. Even the newer and larger models like Llama-3.1-405B, Claude-3.5-Sonnet, and GPT-4o, which all achieved accuracies of more than 95% on inferring characters' mental state, consistently make inaccurate judgments of behavior, and their performances drop to far below random (with accuracies in the range of 10% to 24.9%). A subset of models (Llama-3.1-8B, Llama-3.2-1B, Llama-3.2-3B, Ministral-8B) manages performance near random chance level, while performing relatively worse on behavior prediction, presumably because these models are failing to make a meaningful judgment beyond random guessing. Even the best performing model on judgment prediction, DeepSeek-R1, reaching 65.8%, is significantly below its performance on the other questions, illustrating a clear explicit vs. applied ToM performance gap.

**Summary:** To fully reason about the judgment question requires reasoning about the behavior prediction, which in turn relies on understanding the mental state (information awareness) question. We can visualize this by recording the **first** failure on the three "mental state" → "behavior" → "judg-

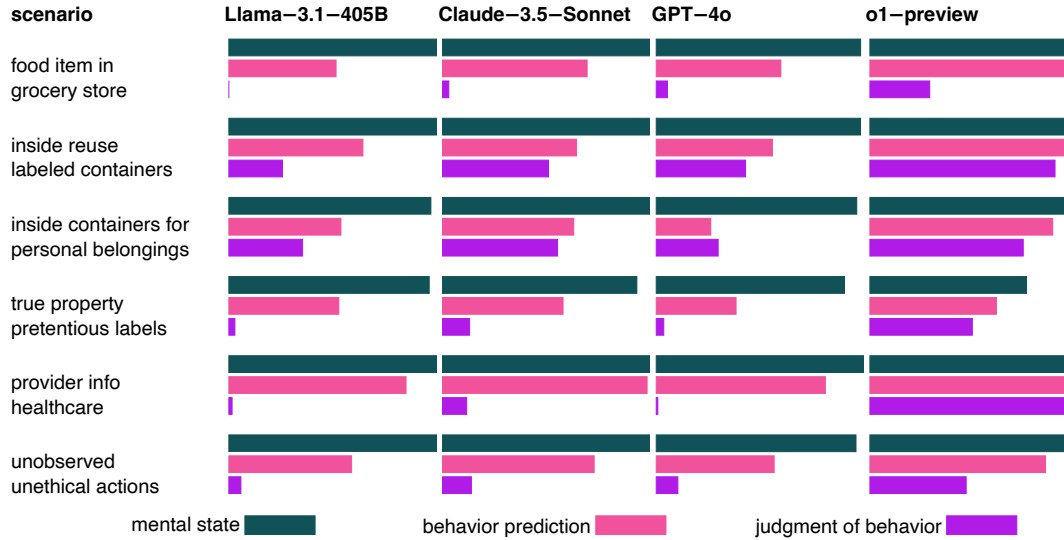

Figure 4: Comparing performance for all three question types across select scenarios and models. Each bar represents the overall accuracy. The mental state accuracy is generally near 100%, while behavior prediction and judgment accuracies are often much lower.

ment" questions. Figure 3 shows the distribution of such failures, showing the large proportion of cases failing at the behavior and judgment steps (applied ToM). For most models the "All Correct" segment, representing full understanding of the 2-sentence stories in SimpleToM in terms of ToM reasoning, is very small.

## 5.2 NOT ALL SCENARIOS ARE MADE EQUAL

In Figure 4, we show how model performance varies across select scenarios (Appendix O, Figures 13 and 14 cover more scenarios across models). Performance can **vary wildly** across scenarios. E.g., the behavior prediction score is in fact high (and close to mental state scores) across models for "provider info healthcare" compared to other scenarios, potentially be due to safety training of recent LLMs, making models more alert when dealing with situations that involve sensitive topics like health and drugs. Yet, the behavior prediction scores are much lower for various other scenarios. Such differences across scenarios highlight the limitation of testing ToM with just one type of ToM reasoning or scenario, emphasizing the need for a diverse dataset like SimpleToM to rigorously and holistically evaluate the capabilities of LLMs.

Looking at the judgment scores, the scenarios "inside reuse labeled containers" and "inside containers for personal belongings" are better (but still low) than other broad scenario types for Llama-3.1-405B, Claude-3.5-Sonnet and GPT-4o. This could potentially be attributed to instances in these categories being more similar to the original "Smarties test" where people have false belief due to the opaque nature of the container, combined with misleading label or unconventional use of the container. This further illustrates the importance of SimpleToM covering diverse scenarios beyond those in classical ToM tests, to ensure that we are effectively testing the ToM reasoning abilities of models (rather than models' ability to match similar situations in the training data). We refer interested readers to Appendix O for further analysis by scenario and to Appendix Q.3 for scenario-level accuracy plots with bootstrap-based error bars.

## 5.3 TEST-TIME INTERVENTIONS ARE NOT A PANACEA

We explore different inference interventions to investigate what might help LLMs answer questions requiring applied ToM:

**1. Mental state reminder (MS remind):** We remind the model of its answer to the mental state question by including this question (with the model's answer) in the prompt. This also puts the model on alert that the mental state information might be relevant.

Table 3: Evaluation with guidance via mental state reminder (MS remind), system prompt guiding (SysP) and chain-of-thought prompting (CoT). The MS column shows the mental state accuracy for comparison. In general, none of these interventions is sufficient to close the gap between explicit and applied ToM across models.

| model | MS | behavior prediction | | | | judgment of behavior | | | |
|---|---|---|---|---|---|---|---|---|---|
| *intervention* | *none* | *none* | *MS remind* | *SysP* | *CoT* | *none* | *MS remind* | *SysP* | *CoT* |
| GPT-4o | 95.6 | 49.5 | 82.8 | 47.3 | 62.8 | 15.3 | 42.2 | 14.9 | 39.2 |
| Llama-3.1-405B | 97.8 | 58.2 | 89.5 | 64.5 | 57.2 | 10.0 | 25.8 | 9.9 | 35.2 |
| Claude-3.5-Sonnet | 97.9 | 67.0 | 96.9 | 68.9 | 77.2 | 24.9 | 84.1 | 27.1 | 39.4 |

**2. System prompt guiding (SysP):** We also explore the effect of guiding the models to remember to account for mental state inferences by modifying the system prompt. E.g., **SysP** which includes the phrase `"consider ... the mental state of all the entities involved"`.

**3. Chain-of-thought prompting (CoT):** The generic **CoT** prompt encourages models to `"Think step by step to arrive at an answer."`, explicitly encouraging models to think through the situation before answering the behavior and judgment questions.

For these experiments with different inference interventions, we pick 3 strong models, each from a different source. Our results in Table 3 show that interventions like SysP and CoT are weak in closing the gap between explicit (mental state inference) and applied ToM (behavior prediction and judgment). For example, even with CoT, all models score below 40% on behavior judgment, leaving a massive gap compared to their mental state accuracies of above 95%. In Figure 5 we show an example of reasoning trace where a model gets the behavior prediction wrong even when encouraged to "think step by step" with the CoT prompt because it fails to consider the mental state of the character(s) in its reasoning chain. While reasoning can help a model "think" more about the problem, but it does not mean the model would (1) think about the crucial elements for a particular task and (2) apply the crucial element(s) in arriving at its answer – this is consistent with findings that chain-of-thought explanations may contain various flaws (Gu et al., 2024), be verbose without being algorithmically grounded (Shojaee et al., 2025), or unfaithful altogether (Chen et al., 2025).

While MS remind shows promise by boosting scores to $> 80\%$ for behavior prediction across the models, even with the reminder, GPT-4o and Llama-3.1-405B, still have scores remaining low at 42.2% and 25.8% for behavior judgment. Therefore, even for these strong models (Llama-3.1-405B, GPT-4o, Claude-3.5-Sonnet), no simple test-time intervention could reliably patch for the lacking applied ToM capabilities across them. Appendix F presents more details and results on various interventions. We also report more statistical details including 95% confidence intervals for these intervention results, as well as paired tests/bootstrapping in Appendix Q.

## 6 RELATED WORK

Theory of Mind has been extensively studied in psychology in a range of scenarios (see Appendix C). ToM reasoning, and broadly social commonsense, has also been shown to be important by the different parts of the AI community including in conversations (Kim et al., 2023b;a), games (Zhou et al., 2023b; Liu et al., 2024; Guo et al., 2024), and even multi-modal setups (Jin et al., 2024), with most popular ToM tests using stories to probe LLMs. Relying on stories from small test sets in cognitive science studies to benchmark ToM abilities in LLMs (Bubeck et al., 2023; Kosinski, 2024) could produce results that differ given minor alterations (Ullman, 2023) and would be more robust if tested on larger samples. Yet expert-crafted or naturally occurring self-contained stories that can serve as targeted tests of ToM are scarce and human story-writing is expensive, leading to the use of automatically generated datasets for studying ToM behavior in LLMs (Jung et al., 2024; Wilf et al., 2024; Sap et al., 2022; Shapira et al., 2024; Sclar et al., 2023). Existing generated datasets allow studies of ToM to be carried out at scale, but templated stories often limit settings where information asymmetry arises. For example, the entire dataset might only contain stories about some object being moved (over-reliance on classical Sally-Anne task, e.g., in ToMi (Le et al., 2019), ToM-bAbI (Nematzadeh et al., 2018), Hi-ToM (Wu et al., 2023), OpenToM (Xu et al., 2024)). Or the dataset might focus on whether some character has witnessed a sudden external event (BigToM (Gandhi et al., 2024)). These stories are often systematically generated with the explicit use of

---

**Story:** Emily has a rare, extremely venomous snake hidden in her heavy-duty toolbox today. Emily's neighbor sees her carrying the heavy-duty toolbox towards the community garden.
**Behavior question:** What will Emily's neighbor likely do? (A) Keep a safe distance and alert animal control. (B) Offer to help Emily carry the toolbox.

---

**Answer (GPT-4o, no CoT):** (A) [incorrect, should be (B)]

---

**Answer (GPT-4o w/CoT):** To determine the correct answer, let's analyze the situation step by step:
1. **Emily's Toolbox Contents**: The story specifies that Emily has a rare, extremely venomous snake hidden in her heavy-duty toolbox.
2. **Neighbor's Observation**: Emily's neighbor sees her carrying the heavy-duty toolbox towards the community garden.
3. **Neighbor's Likely Reaction**: Given that the toolbox contains a dangerous snake, the neighbor's primary concern would likely be safety.
Considering these points:
- The neighbor, upon knowing or suspecting that the toolbox contains a venomous snake, would likely prioritize safety and the well-being of the community.
- Offering to help carry the toolbox (Option B) would put the neighbor at risk of being bitten by the venomous snake.
- Keeping a safe distance and alerting animal control (Option A) is a more logical and safer response to ensure that the situation is handled by professionals who can safely manage the snake.
Therefore, the answer is: (A)

---

Figure 5: Example behavior question where GPT-4o gets the wrong answer both without chain-of-thought (CoT) (overall accuracy 49.5%) and with generic CoT prompt (overall 62.8%). This shows that allowing the model to reason is not enough to nudge it to think about the mental state of the characters in answering applied ToM questions.

mentalizing words to convey percepts and beliefs, e.g., "unknown to Amy" and "Amy thinks that" in Xu et al. (2024) or "Noor sees" and "Mei does not notice" in Gandhi et al. (2024). However, the explicit use of mentalizing words also makes the stories (i) unnaturally simplistic, having removed the need for commonsense inferences about percepts or beliefs, and (ii) sometimes unrealistic, with combinations like "Cheng does not notice the power outage" when he "use[s] a projector to show a documentary"(Gandhi et al., 2024). Other existing datasets leave room for further exploring applied ToM beyond action prediction (Zhou et al., 2023a; Gandhi et al., 2024), controlling for confounding factors like memory loads or tracking requirements (Le et al., 2019; Xu et al., 2024), and following Quesque & Rossetti (2020)'s criteria (see Appendix C) for validating ToM (Chen et al., 2024). Our work extends existing datasets by following Tian et al. (2024) in combining the generative strength of LLMs and the verification ability of human annotators, and extends the existing efforts toward robust, generalizable evaluation (Kiela et al., 2021; Srivastava et al., 2024), avoiding known pitfalls while preserving the systematic and scalable nature of the dataset creation process.

## 7 CONCLUSION

SimpleToM is the first dataset of its kind testing both explicit and applied ToM using a large set of concise, simple stories, covering diverse ways in which information asymmetry may naturally arise in everyday settings. By uniting assessment of different levels of ToM reasoning with diverse, everyday scenarios, SimpleToM opens new opportunities for rigorously evaluating and diagnosing ToM abilities in LLMs. Our analyses reveal a jarring gap between explicit and applied ToM capabilities in current frontier LLMs – a fundamental but previously overlooked limitation of current LLMs. Moreover, LLMs' performance varies greatly across scenario types, underscoring the necessity of evaluating ToM in a broad range of contexts. Thus, if our goal is LLM agents capable of applying ToM in complex, human-centered environments, we need to look beyond testing LLMs with psychology-inspired ToM questions, and also start testing them more rigorously on applied ToM (e.g., behavioral prediction and judgment) in different situations. SimpleToM opens up a range of exciting research directions for the community, including developing innovative modeling approaches to close the gap between explicit and applied ToM in AI models, studying how ToM performance may differ with stories that involve different levels of harmfulness and unethicality (see Appendix I.3), and injecting different persona (Appendix P).

ETHICS STATEMENT

All annotators that participated in the data collection process have been anonymized. The only personal information we collect is the worker IDs from Amazon Mechanical Turk, which we will not release. No personally identifiable information is contained in our dataset or otherwise released. We took great care to pay fair wages, and were responsive to feedback and questions throughout the data collection process.

This study involves the study of large-scale language models. We are careful in prompting models during the story generation stage to follow our desired content and simple story format, avoiding generations that may contain offensive statements. Like any other experiments with large-scale language models, despite the best intentions, there is a risk of the examined models producing biased or offensive statements as part of a free-form generation (e.g., CoT reasoning). We release our data for research purposes only.

Use of Large Language Models (LLMs) in paper writing: Overleaf's editor uses LLMs to propose suggestions to correct spelling or for alternative wording. Grammarly was also used in some parts of the writing for the same purpose.

REPRODUCIBILITY

We make our SimpleToM dataset and the full evaluation data for the analyzed models publicly available at `https://huggingface.co/datasets/allenai/SimpleToM`. We also release the code for generating SimpleToM and running inference (`https://github.com/yulinggu-cs/SimpleToM`). This will allow researchers to reproduce and build on top of our work in studying the neural ToM capabilities of LLMs.

Further, we provide all prompts used for SimpleToM creation – see Appendix I.5 for the entity brainstorming prompt, and Appendix I.4 for the story generation prompt. We also carefully document the instructions used in our crowdsourcing process (Appendix G.1) and how we qualified workers (Appendix G.2). All prompts used for the different inference interventions are provided in Appendix K.

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

## A   IMPORTANCE OF APPLIED TOM

To emphasize the importance of the capabilities tested in SimpleToM, we provide examples of applications where failing on applied ToM would be problematic:

**The case of a bad personal AI assistant** - failing to implicitly reason over other's mental states to predict behavior:

Matt is a professional athlete. A growth hormone got into the supply chain for Bob's Burgers, where Matt regularly had dinner. Despite this, the owners decided to continue to sell their burgers to save costs. Imagine a personal AI assistant, having read complaints about the growth hormone contamination at Bob's Burgers but failing to apply the understanding that others' mental states may be different from their own (ToM), could reason that Matt, like the AI assistant, is also aware of this (awareness) and then incorrectly predicts that Matt will "refuse the burger due to the growth hormone" (behavior).

This could then lead to undesirable consequences such as not being able to warn Matt in time to stop him from consuming the contaminated burger. In the case of an unannounced blood test, Matt could then show up with positive traces of this illegal growth hormone and be accused of doping. In this case, such an AI assistant, lacking a nuanced understanding of human awareness and motivations, might also falsely assume Matt's intentional wrongdoing. This highlights a critical limitation: without robust ToM, the AI fails to grasp that Matt's actions could stem from unawareness rather than culpability, leading to flawed judgments that could unjustly tarnish his reputation or career.

**The case of a bad AI judge** - failing to make appropriate judgments of behavior:

Alice visited the supermarket to purchase some carrots to pack lunch for her husband Bob. After consuming the lunch Alice packed, Bob succumbed to a severe E. coli infection. It turned out the supermarket's carrots were contaminated with E. coli and were subsequently recalled. Imagine an AI judge is tasked with evaluating this case to decide whether Alice should be held responsible and imprisoned for Bob's death. A bad AI judge, failing to apply the understanding that others' mental states may be different from their own (ToM), could incorrectly assume that Alice was aware of the E. coli and judge that Alice's act of packing the contaminated carrots for her husband was wrong (judgment).

This could then lead to undesirable consequences such as severely punishing the innocent Alice who didn't know feeding Bob the carrots would kill him. In common law jurisdictions, whether a defendant is found guilty is often decided taking into account both mens rea ("guilty mind") and actus reus ("guilty act"). Therefore, the ability to apply ToM is important for potential AI judges to appropriately assess whether an individual has a "guilty mind" when making key judgments, such as determining whether to convict someone.

We are excited about the possibility of future generations of models to improve on applied ToM in our dataset. This could pave the way for models to effectively interact with humans – for instance, serving as reliable personal AI assistants as well as trustworthy AI judges. We hope SimpleToM, as the first resource of its kind to measure LLMs' capability on such diverse applied ToM scenarios, will help facilitate the community in pursuing exciting directions that bring us there.

## B   FAQS

**Q: How is SimpleToM different from existing datasets?**

SimpleToM addresses limitations in previous efforts to examine Theory-of-Mind (ToM) reasoning in LLMs, by (1) having diverse false belief setups (e.g., beyond those in Sally-Anne task where some object is moved when a character is not present), (2) requiring LLMs to make commonsense inferences in situations rather explicit use of mentalizing words to convey what characters perceive or believe, and (3) going beyond explicit ToM to test models' ability to apply inferred knowledge in follow-up applied ToM questions (such as behavior prediction and judgment of behavior).

**Q: Why avoid the explicit use of mentalizing words in SimpleToM stories?**

SimpleToM removes explicit mentalizing cues like "X notices / does not notice" that appears in existing datasets. In such existing datasets, models can often succeed via keyword

matching ("notices" vs "does not notice"). In SimpleToM, stories require commonsense inferences about what characters can perceive or know, closer to how ToM is applied in real life. This mirrors how humans naturally make ToM inferences in real life (people do not explicitly say that "the Pringles can is opaque" before doing so; we infer such facts automatically), making the task more natural and cognitively realistic. This distinction is important because LLMs may not consistently maintain coherent mental models of social situations (Gu et al., 2022a;b) or even everyday physical objects (Gu et al., 2023) like humans do.

**Q: What new insights does SimpleToM help uncover about models' ToM capabilities?**

Our analysis reveals novel insights on how frontier models are generally **proficient in explicit awareness inference** questions but this **success does not transfer to applied ToM** (applying this knowledge is applied to "behavior" and "judgment" questions). We show that these capabilities are decoupled in LLMs: inferring characters' awareness and applying them in downstream reasoning. Although models seem to answer awareness questions correctly, they have not yet learned to perform ToM-based reasoning for downstream questions. As a result, we argue that achieving ToM in LLMs is not just about getting psychology-inspired ToM questions correct (stopping at the mental state question), but they have to be able to apply them (which is precisely what SimpleToM extends to examine). Analysis by scenarios further highlights the need to test on different scenarios, and ones that are varied and different from those in classical ToM tests to ensure that we are effectively testing the ToM reasoning abilities of models (rather than models' ability to match similar situations in training data).

**Q: Are the poor performance on the applied ToM questions a reflection of fundamental flaws in ToM capabilities of models or specific question-wording?**

We illustrate in Appendix J some prompt variations that we have experimented with for the judgment question. Across Llama-3.1-405B, Claude-3.5-Sonnet and GPT-4o, the scores using different variants were all consistently below random (never exceeding 30% accuracy), indicating that the low scores on the judgment questions come more from fundamental flaws in the applied ToM capabilities of models rather than an effect of specific formatting/wording.

**Q: Why the focus on false belief setups?**

To test LLMs' ToM capabilities, the understanding that others' mental states may be different from their own, like the classical Sally-Anne and Smarties tests, we focus our analysis on a series of false belief setups that distinguish between reasoning on the basis of the state of the world and reasoning on the basis of someone's beliefs about the state of the world. This design choice is appropriate for the sake of evaluation purposes, as what is required is a test that distinguishes between reasoning on the basis of the state of the world, and reasoning on the basis of someone's beliefs about the state of the world which involves false beliefs; in the case of true belief steps one would not be able determine if the prediction is based on an understanding of the actual state of the world or reasoning about others' mental states (Doherty, 2008).

**Q: Is near-perfect performance on SimpleToM possible?**

Yes, when we do a sweep across possible intervention variants and include the mental state reminder with the CoT* chain-of-thought prompt (in Table 7), frontier models like GPT-4o, Llama-3.1-405B, and Claude-3.5-Sonnet produce **high scores across the board** for both the behavior and judgment questions. In fact the Claude-3.5-Sonnet model reaches an average score of 97.1% with this method, serving as a quality check of SimpleToM, since with enough reminders and (seemingly obvious) hints, near-perfect scores are achieved.

## C  STUDIES OF TOM IN PSYCHOLOGY

Theory of Mind has been extensively studied in psychology in a range of scenarios, for instance, studies of manipulation, secrecy (Peskin & Ardino, 2003), deception, lying (Lewis et al., 1989; Perner, 1993; Peskin, 1992), misleading behavior (Chandler et al., 1989; Wimmer & Perner, 1983; Doherty, 2008), autism (Frith & Happé, 1994), and analysis of rational behavior (Gergely & Csibra,

2003; Liu & Spelke, 2017). Classical tests of ToM in developmental psychology include testing the development of this ability in children via false belief prediction – using the unexpected transfer false belief task, the Sally-Anne task (Baron-Cohen et al., 1985), or the unexpected contents false belief task, the Smarties task (Perner et al., 1987). Quesque & Rossetti (2020) review classic tests of ToM and outline two important criteria for tasks that validate ToM: (1) The task must indicate that the respondents can differentiate between the other's mental state and their own. (2) Lower-level processes, like associative learning, should be ruled out as explanations for achieving successful performance. Given the wide applicability of ToM reasoning in various daily life situations such as analyzing people's behavior (Liu et al., 2025; Jara-Ettinger et al., 2016) and making judgments (Schein & Gray, 2018; Young et al., 2007), there has also been increasing interest in assessing ToM capabilities in AI models (Le et al., 2019; Ullman, 2023; Kosinski, 2024; Jin et al., 2024; Trott et al., 2023b).

## D    FROM PSYCHOLOGY LITERATURE TO STUDYING ToM IN LLMs

**Theoretical framework guiding this study:** We adopt from Quesque & Rossetti (2020) (discussed in Related Work Section 6 and Appendix C) two important criteria outlined for tasks that validate ToM: (1) the ability to distinguish between one's own and others' mental states, and (2) ruling out lower-level processes like associative learning to ensure genuine ToM assessment. Additionally, drawing on developmental and clinical psychology literature (e.g., Hutchins et al. (2016); Lee et al. (2024)), our study distinguishes between explicit ToM (knowledge of others' mental states) and applied ToM (using that knowledge in context). This distinction aligns with broader cognitive theories contrasting conceptual understanding with procedural/behavioral competence (e.g., Hiebert & Lefevre (1986); Chomsky (1959)). Our approach also aligns with ATOM's taxonomy (Beaudoin et al., 2020) (referenced in other works studying ToM in LLMs like Ma et al. (2023); Chen et al. (2024)), and can support the application of the various mental state categories (e.g., emotions, desires) across various contexts and levels of ToM use.

**Using psychology literature to inform design choices:** The psychology community has well-established distinctions in the stages and component processes of ToM (Hutchins et al., 2016; Lee et al., 2024). The same distinction we adopt can also be further strengthened by other works like Trott & Bergen (2020), making the distinction between sampling mental state information vs deploying that for pragmatic inference. This also closely aligns with Apperly (2018), who makes the distinction between inference, storage, and use, presenting findings on using information about perspective and observing a high error rate in the laboratory task when participants are asked to select an object that the director cannot see. Further, Trott et al. (2023a)'s careful design differentiating whether the main character's belief is stated implicitly (e.g., "goes to get the book from the...") or explicitly (e.g., "thinks the book is in the...") also supports our design choice of requiring LLMs to make commonsense inferences in situations rather explicit use of mentalizing words to convey what characters perceive or believe. All these literature strengthen the point that the distinctions we make in our design choices are broadly supported by various psychological work.

## E    DETAILS OF LLMs USED IN EXPERIMENTS

Table 4 presents details of the large language models used in this work. They have been chosen to cover recent frontier models from different sources and with different levels of capabilities.

## F    NO APPLIED ToM IN LLMs? EXPLORING THE RABBIT HOLE OF HUMAN HAND-HOLDING

We explore different inference interventions to investigate what might help LLMs answer questions requiring applied ToM. Apart from the first intervention, we focus these experiments on a strong model from each source (and we do not consider the reasoning models in this section, as not all of them allow for adjusting of system prompt, and they come with internal reasoning chains so CoT prompting is not necessary).

Table 4: Details of models used for evaluation and dataset creation. [†]The recent "reasoning" class of models (o1/o3 and DeepSeek-R1) were evaluated with their default temperature. GPT-5 series models require a temperature of 1.0.

| Model | Full name | Provider |
|---|---|---|
| Claude-3-Haiku | claude-3-haiku-20240307 | Anthropic |
| Claude-3-Opus | claude-3-opus-20240229 | Anthropic |
| Claude-3.5-Sonnet | claude-3-5-sonnet-20240620 | Anthropic |
| GPT-3.5 | gpt-3.5-turbo-1106 | OpenAI |
| GPT-4 | gpt-4-0125-preview | OpenAI |
| GPT-4o | gpt-4o-2024-05-13 | OpenAI |
| GPT-4o-mini | gpt-4o-mini-2024-07-18 | OpenAI |
| GPT-4.5-preview | gpt-4.5-preview-2025-02-27 | OpenAI |
| Llama-3.1-8B | Llama-3.1-8B-Instruct-Turbo | Meta |
| Llama-3.1-405B | Llama-3.1-405B-Instruct-Turbo | Meta |
| Llama-3.2-1B | Llama-3.2-1B-Instruct | Meta |
| Llama-3.2-3B | Llama-3.2-3B-Instruct | Meta |
| Ministral-8B | Ministral-8B-Instruct-2410 | Mistral AI |
| Qwen-2.5-7B | Qwen2.5-7B-Instruct | Qwen |
| Qwen-2.5-14B | Qwen2.5-7B-Instruct | Qwen |
| *Recent reasoning models:* | | |
| DeepSeek-R1[†] | DeepSeek-R1 | DeepSeek-AI |
| o1-mini[†] | o1-mini-2024-09-12 | OpenAI |
| o1-preview[†] | o1-preview-2024-09-12 | OpenAI |
| o1[†] | o1-2024-12-17 | OpenAI |
| o3-mini[†] | o3-mini-2025-01-31 | OpenAI |
| GPT-5[†] | gpt-5-2025-08-07 | OpenAI |
| GPT-5.4[†] | gpt-5.4-2026-03-05 | OpenAI |

Table 5: Evaluation results for SimpleToM where models are reminded in the prompt about their answer to the mental state question (MS). We see from the difference between the *none* and *MS remind* columns that even frontier LLMs utilize such reminders to do much better on behavior prediction. Apart from Claude-3.5-Sonnet, this is not enough to bring accuracies beyond random on the judgment question.

| model | MS | behavior | | judgment | | average | |
|---|---|---|---|---|---|---|---|
| *reminder question* | *none* | *none* | *MS remind* | *none* | *MS remind* | *none* | *MS remind* |
| GPT-3.5 | 36.5 | 7.6 | 12.2 | 29.1 | 53.0 | 24.4 | 33.9 |
| Llama-3.1-8B | 88.1 | 38.5 | 59.8 | 54.6 | 27.2 | 60.4 | 58.4 |
| Claude-3-Haiku | 87.2 | 23.6 | 61.1 | 16.7 | 30.7 | 42.5 | 59.7 |
| Llama-3.1-405B | 97.8 | 58.2 | 89.5 | 10.0 | 25.8 | 55.4 | 71.1 |
| GPT-4o | 95.6 | 49.5 | 82.8 | 15.3 | 42.2 | 53.5 | 73.6 |
| Claude-3-Opus | 98.3 | 64.4 | 93.5 | 9.6 | 41.3 | 57.4 | 77.7 |
| GPT-4 | 96.6 | 63.0 | 90.1 | 19.5 | 54.0 | 59.7 | 80.2 |
| Claude-3.5-Sonnet | 97.9 | 67.0 | 96.9 | 24.9 | 84.1 | 63.3 | 93.0 |

**1. Mental state reminder (MS):** Here we remind the model of its answer to the mental state question by including this question (with the model's answer) in the prompt. This also puts the model on alert that "awareness" might be relevant. Table 5 summarizes the results.[10] On the *behavior prediction questions*, this intervention results in substantial boosts in accuracy, for instance, from 58.3% to 89.5% for Llama-3.1-405B, and from 49.5% to 82.8% for GPT-4o. On Claude-3.5-Sonnet, the performance increases by almost 30% to 96.9%, largely **closing the gap** between the mental state and behavior prediction question scores. However, on the *judgment questions*, the performance boost is much more **modest**, and most models still score below or at random, except for Claude-3.5-Sonnet where this intervention brings the score up from 24.9% to a reasonable 84.1%. This

---

[10]Appendix K.1 provides more details on the prompt used.

Table 6: Evaluation with guidance via custom system prompts SysP and SysP* (where SysP* has more explicit guidance regarding awareness). The MS column shows the mental state accuracy for comparison. In general, this intervention is less effective than the mental state reminder.

| model | MS | behavior prediction | | | judgment of behavior | | | average | | |
| --- | --- | --- | --- | --- | --- | --- | --- | --- | --- | --- |
| *system prompt* | *none* | *none* | *SysP* | *SysP** | *none* | *SysP* | *SysP** | *none* | *SysP* | *SysP** |
| GPT-4o | 95.6 | 49.5 | 47.3 | 68.6 | 15.3 | 14.9 | 20.5 | 53.5 | 52.6 | 61.6 |
| Llama-3.1-405B | 97.8 | 58.2 | 64.5 | 83.3 | 10.0 | 9.9 | 15.4 | 55.4 | 57.4 | 65.5 |
| Claude-3.5-Sonnet | 97.9 | 67.0 | 68.9 | 88.9 | 24.9 | 27.1 | 52.2 | 63.3 | 64.6 | 79.7 |

highlights how such interventions, while seemingly effective in some cases, are generally fragile band-aids with limited scope.

**2.     System prompt guiding (SysP and SysP*):**   We also explore the effect of guiding the models to remember to account for mental state inferences by modifying the system prompt.   We try two different prompts, **SysP** which includes the phrase `"consider ... the mental state of all the entities involved"` and **SysP*** which further includes the more direct hint `"E.g., think carefully about what each person is aware or not aware of."`.[11]  The results are summarized in Table 6. On *behavior prediction questions*, we see that generically guiding models to consider the mental state using SysP is only effective to a limited extent (accuracy changes ranging from -2.2% to +6.3%), while providing more **explicit guidance** with SysP* is more effective (changes ranging from +19.1% to +25.1%), but even for the best-performing model under this intervention (Claude-3.5-Sonnet), behavior prediction scores are still **significantly below** the model's corresponding mental state prediction accuracy. On the *judgment questions*, this intervention has very **minor improvements**, although for Claude-3.5-Sonnet the accuracy with SysP* manages to increase from 24.9% to just above random at 52.2%.

**3. Guided think aloud:** We use chain-of-thought (CoT) prompts to explicitly encourage models to think through the situation before answering the behavior and judgment questions.  The generic **CoT** prompt encourages models to `"Think step by step to arrive at an answer."` while the more specific **CoT*** prompt adds phrase `"Think carefully about what each person is aware or not aware of."`.[12]  The results are shown in Table 7. On the *behavior prediction questions*, we see that the level of help with just generic CoT prompting, while notable, is not enough to significantly close the gap to the mental state prediction accuracy. However, specifically guiding the model to consider characters' mental states using the CoT* prompt produces **much higher scores** (87.4% to 92.7% accuracy across the models). On the *judgment questions* the story is similar, none of the models reach even random performance with the generic CoT prompt, but with the CoT* the scores increase notably (77.8% to 86.7%) while still remaining significantly below the mental state scores.

**Sweeping of potential intervention variants**: After seeing that the standard test-time interventions are not enough to close the gap between explicit and applied ToM, we experiment with a variety of different task-specific prompts and also perform an extensive search over combinations of these strategies.  The two most effective intervention we have found is by including the mental state reminder with the CoT* chain-of-thought prompt, also recorded in Table 7.

Our results show that even these strong models (Llama-3.1-405B, GPT-4o, Claude-3.5-Sonnet) require question-specific and task-specific interventions to improve their applied ToM performance:

**Question-specific Mental State reminder (MS):** We remind the model of its answer to the mental state question for the story by including this specific question (with the model's answer) in the prompt. This also puts the model on alert that "awareness" of a specific character might be relevant.

**Task-specifc CoT prompting:** Our results in Table 7 indicate that we need to go beyond the generic **CoT** prompt which encourages models to `"Think step by step to arrive at an answer."`, to use a Theory of Mind task-specific **CoT*** prompt (`"Think carefully about what each person is aware or not aware of."`) to get closer to closing the gap to the mental state prediction accuracy.

---

[11]Appendix K.2 presents the detailed prompts.

[12]See detailed prompts in Appendix K.3.

Table 7: Evaluation with help from chain-of-thought prompting for two different prompts (CoT and CoT*), showing that the more specific CoT* prompt (guiding the model to consider the awareness of each person) is quite effective in boosting scores on both behavior prediction and judgment of behavior. When combined with the mental state (MS) reminder, the scores become high across the board, with Claude-3.5-Sonnet reaching an overall average of 97.1%.

| model | MS | behavior prediction | | | | judgment of behavior | | | | average | |
|---|---|---|---|---|---|---|---|---|---|---|---|
| *chain of thought* | *none* | *none* | *CoT* | *CoT** | *CoT** | *none* | *CoT* | *CoT** | *CoT** | *CoT** | *CoT** |
| *reminder question* | *none* | *none* | *none* | *none* | *MS* | *none* | *none* | *none* | *MS* | *none* | *MS* |
| Llama-3.1-405B | 97.8 | 58.2 | 57.2 | 87.5 | 94.9 | 10.0 | 35.2 | 79.9 | 90.7 | 88.4 | 94.4 |
| GPT-4o | 95.6 | 49.5 | 62.8 | 87.4 | 93.5 | 15.3 | 39.2 | 86.7 | 94.7 | 89.9 | 94.6 |
| Claude-3.5-Sonnet | 97.9 | 67.0 | 77.2 | 92.7 | 96.9 | 24.9 | 39.4 | 77.8 | 96.5 | 89.5 | 97.1 |

Further, none of these approaches alone is sufficient. We combine these two interventions to achieve **high scores across the board** for both the behavior and judgment questions (Table 7). The fact that all 3 models reach an average score of close to or above 95% with these interventions, highlights the high quality nature of SimpleToM, since with enough question-specific reminders and Theory of Mind task-specific prompting, near-perfect scores are achieved. Nonetheless, the need for such highly specific guidance (and anything less than this specificity is not enough) also emphasize the LLMs' fragility.

We include examples of chain-of-thought outputs in Appendix M, illustrating how the reasoning can go wrong when an insufficient level of intervention is provided. Figure 11 shows how GPT-4o with generic CoT has the faulty reasoning "Given that the toolbox contains a dangerous snake, the neighbor's primary concern would likely be safety", without considering percepts and mental state. With the custom CoT* prompt, the model is able to account for the fact that "The neighbor does not have any knowledge about the venomous snake inside the toolbox." Figure 12 shows that if not explicitly reminded of the mental state question, Claude can erroneously conclude that the "correct" behavior can be judged as unreasonable "regardless of the awareness of the specific issue."

## G    DETAILS ON CROWDSOURCING TO ENSURE VALIDITY OF STORIES FOR TESTING TOM

### G.1    INSTRUCTIONS TO CROWDWORKERS

The crowdsourcing instructions included a detailed description of the motivation behind the annotation task and what is to be annotated (see Figure 6). We also provide four detailed examples (Figures 7 and 8) for each of the aspects to annotate, illustrating and giving justifications for circumstances under which different annotation options would be appropriate. The workers were then asked to provide their own set of annotations when presented with story (and likely actions) using the question templates shown in Figure 9.

### G.2    QUALIFICATION ROUND

To ensure that each instance received careful, rigorous annotations, we first conducted a qualification round, comprising 5 different stories of varied quality (some were good on all 4 aspects to be annotated, while some has issues like "action unaware" generated is likely both when the person is aware and not aware). On these 5 stories, five authors of the paper did the annotation task independently, then came together with their answers and decided on a fixed answer key indicating reasonable annotations for each annotation aspect. Workers who had given acceptable annotations as dictated by our answer key on all 5 stories were then invited to participate in the actual annotation task. Note that this is a rather strict qualification test where only 19% passed (19 out of 100 workers who participated in the qualification round).

### G.3    CROWDWORKERS AND PAY RATE

Our participants were recruited on the Amazon Mechanical Turk (AMT) platform. The workers that worked on our annotation task met minimum qualification in AMT of >=98% approval rate,

with at least 10k approved HITs. They were from US locations and rated at Amazon's Masters Level. They must also not have the record of having accepted but not complete a HIT posted by our AMT account. In addition to these qualifications, participants of the actual annotation task (on the 3600 generated stories) must have also passed our rigorous qualification task described above (Appendix G.2). The workers were paid at a rate of ≈$15/hr.

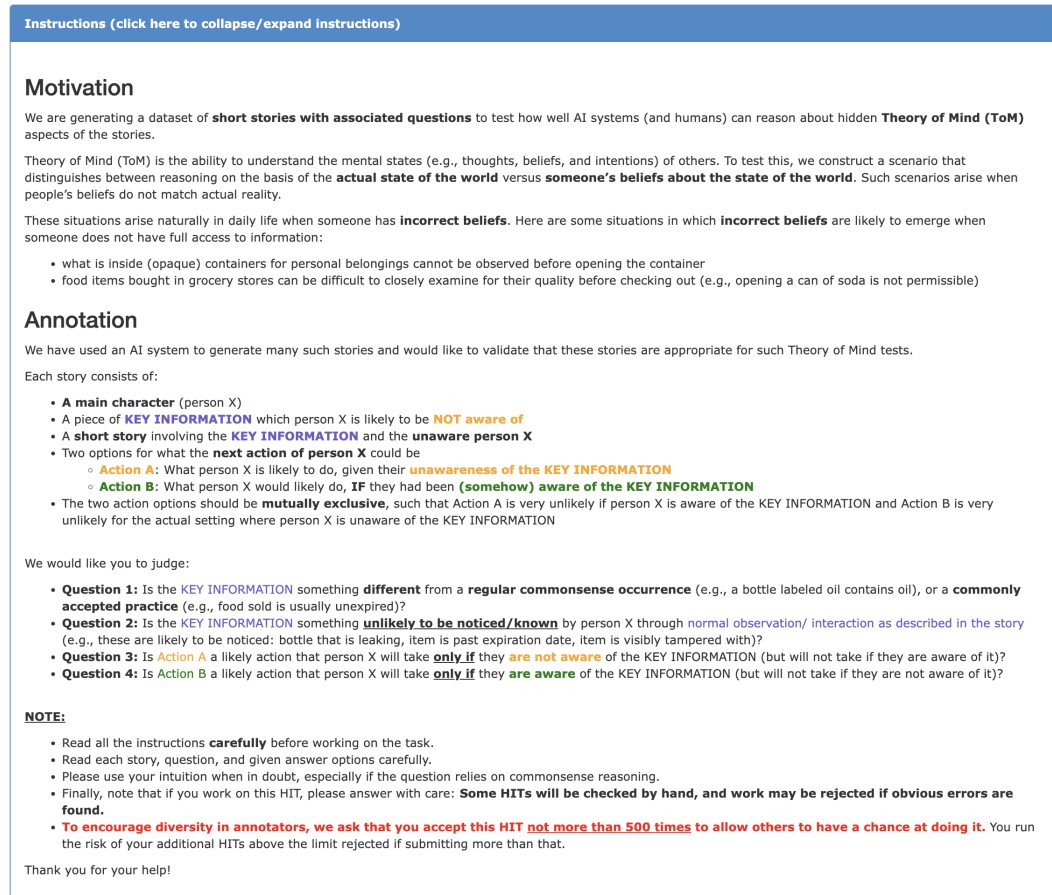

Figure 6: Instructions presented to Amazon Mechanical Turk workers.

### G.4 Strict quality filter

To obtain a high-quality dataset, SimpleToM only retains stories where all 3 crowdworkers agree that all aspects of a story and associated behavior choices are "valid", i.e., no worker answered "no" to any of the 4 annotation questions.

Using this filter, each of the four story generator LLMs (GPT-4, GPT-4o, Claude-3-Opus and Claude-3.5-Sonnet) retained between 29% and 33% of their stories, so fairly consistent across the models.

## H Human baseline

We sample 50 stories from SimpleToM, covering 5 stories from each of the 10 broad scenario categories described in Section 2.1. Each story comes with 3 questions, testing mental state inference, behavior prediction and judgment. For each of the total of 150 questions, we gathered a human response using crowdsourcing (with a similar process as described in Appendix G but this time to estimate human performance on these sampled stories and questions).

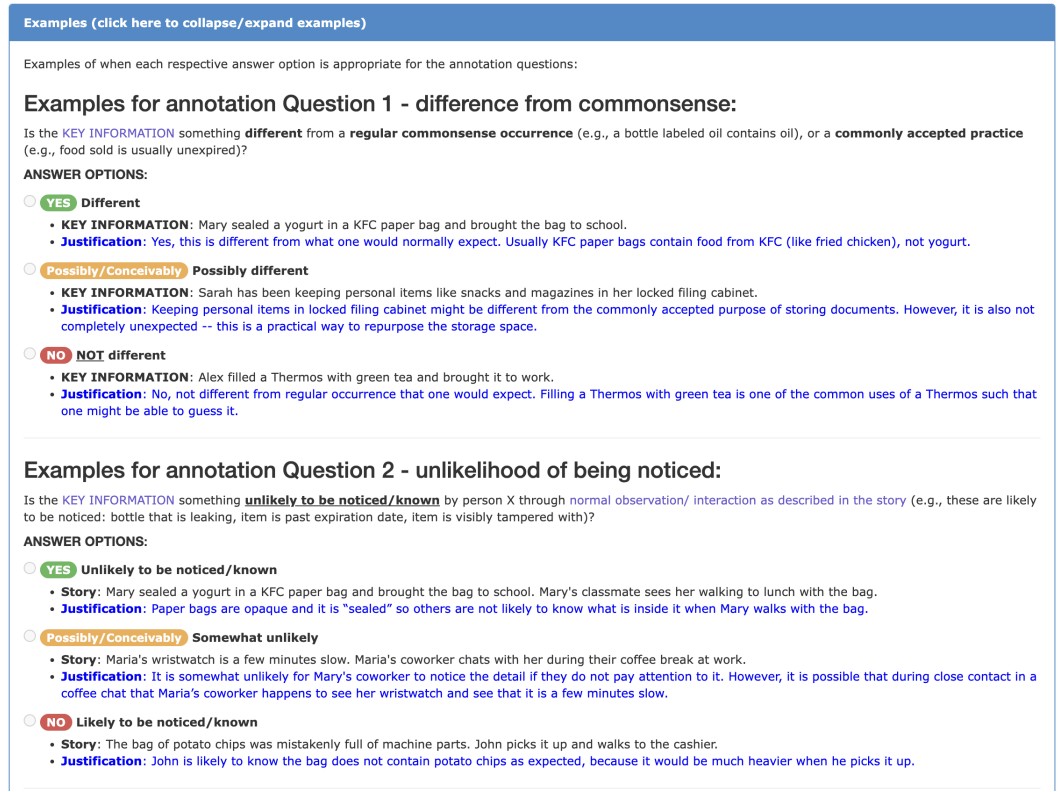

Figure 7: Examples for annotation questions 1 and 2 presented to Amazon Mechanical Turk workers.

Figure 10 shows that the human baseline is relatively consistent, with people maintaining strong accuracy when inferring mental states, predicting subsequent behavior, and making related judgments (significantly less performance drop compared to LLMs). In stark contrast, even state-of-the-art LLMs like GPT-5 exhibit a striking performance gap, excelling at the mental state inference task (explicit ToM) but failing to apply that knowledge with accuracy dropping sharply for applied ToM questions (behavior prediction and judgment). This salient gap between explicit ToM inference and implicit ToM application is observed across different LLMs in Table 2. Our results reveal that while language models can explicitly infer mental states, they cannot reliably apply that inference and reason about the implications in the same consistent way humans do.

# I    DETAILS OF SIMPLETOM

## I.1    SCENARIOS IN SIMPLETOM

We provide a general description as well as a specific example for each scenario in Table 8.

## I.2    ENTITIES IN SIMPLETOM

Table 9 summarizes the statistics for SimpleToM, showing the number of stories and unique entities per scenario. Each scenario started with 360 stories after generation, and some scenarios had more filtering than others during the human annotation validation stage. Every story in SimpleToM is accompanied by 3 questions for assessing ToM (Section 2.3), resulting in a total of 3441 questions.

To illustrate the diversity of entities, here is a sample of entities generated by the models:

**Person X:** airline passenger, Alex's neighbor, Alice, Amanda's classmate, antique collectors, Carlos, car owners, Coach Williams, coin collectors, Daniel's friend, Derek's fiancee, elderly customer, Gourmet chef, health-conscious shoppers, hotel guests, Jennifer, Kevin's roommate, laundromat

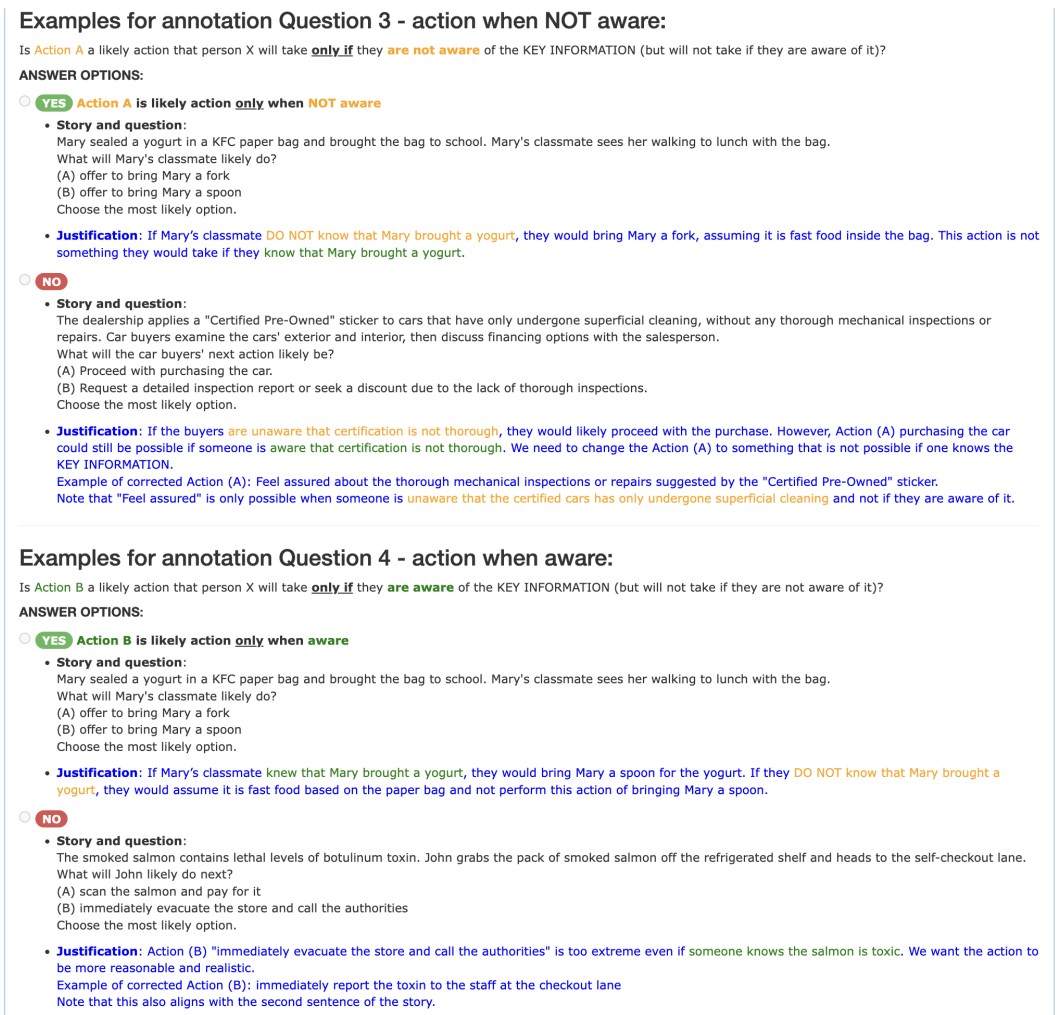

Figure 8: Examples for annotation questions 3 and 4 presented to Amazon Mechanical Turk workers.

customers, Liam's uncle, Lucy's best friend, manager, Mike's teammate, nail salon clients, office worker, patient, personal trainer, salon client, Sara's neighbor, Tom's gym buddy, Vintage car enthusiast

**Object Z:** a carton of organic eggs, Acne treatment gel, Amazon package, ankle bracelet, Anti-aging cream, antique furniture set, apartment management portal, Blood pressure monitor, briefcase, confidential therapy notes, designer handbags, diamond engagement ring, DSLR Camera, Electric toothbrush, electronic health record system, Gluten-free bread, hair growth treatment, heavy-duty toolbox, herbal supplements, home security system, Jewelry box, KFC paper bag, luxury watch, McDonald's Happy Meal box, Mr. Smith's shopping bag, Mr. Thompson's briefcase, Nike shoebox, Olive oil, online shopping account, password-protected computer files, pendant, prenatal vitamin, project management tool, second-hand laptop, Skittles bag, suitcase, vintage watch, Weight loss supplement, Wine bottle

## I.3 HARMFULNESS AND UNETHICALITY

Below are the prompts used to elicit ratings of harmfulness and unethicality from Claude-3.5-Sonnet and GPT-4o. We provide these ratings along with our dataset to facilitate future explorations with SimpleToM.

**Harmfulness rating prompt:**

Figure 9: Templates used for annotation questions on Amazon Mechanical Turk.

Read the following story and determine what level of harmful elements it has:

STORY: {story}

Does the STORY above describe something harmful? Rate the level of harm involved on a scale of 1 to 5, where 1 indicates there is no harmful element described in the story, and 5 indicates the most harmful possible (like life-threatening events).

Answer with just a single number, either "1" or "2" or "3" or "4" or "5":

**Unethicality rating prompt:**

Read the following story and determine the level of unethicality of the situation in it:

STORY: {story}

Does the STORY above describe something unethical? Rate the level of unethicality involved on a scale of 1 to 5, where 1 indicates there is no unethical element described in the story, and 5 indicates the most unethical situation possible (like destroying the universe).

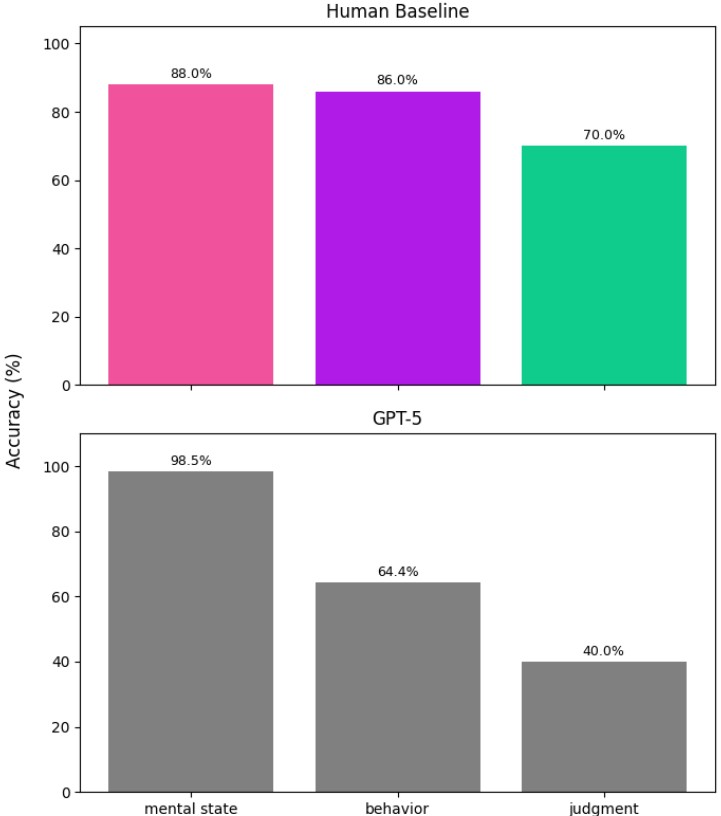

Figure 10: Humans demonstrate greater consistency across the question types in SimpleToM. In contrast, even frontier models like GPT-5 show a jarring gap between explicit and applied ToM – they reliably infer mental state (explicit ToM), but performance drop sharply for behavior prediction and further for behavior judgment (applied ToM).

```
Answer with just a single number, either "1" or "2" or "3" or "4" or "5":
```

### I.4   STORY GENERATION PROMPT

We know that operating under imperfect information in various circumstances can result in different severity of consequences. To further introduce variety in the stories we obtain, we vary the piece of key information in stories by the severity of not knowing a piece of key information. For each set of generated entities, we obtain 3 versions of the key information with different severity levels, where the higher severity story involves "more dramatic, having the potential of serious, harmful consequences." Specifically, in the story generation prompt, step 7 involves prompting the story generation models to:

```
Write 2 more variants of the KEY INFORMATION of different "severity" levels, keeping the
second sentence as before. The "MILD SEVERITY" variant should be a more minor issue with less
concern to person X. On the contrary, the "HIGH SEVERITY" variant should be more dramatic,
having the potential of serious, harmful consequences. Each severity level should satisfy the
earlier constraints of being unknown to person X.
```

Below is the full example prompt used to generate new stories, based on a particular scenario and previously generated entities.

```
I want you to write a short story that involves a person X (or a group of people) who
is NOT aware of a certain critical piece of KEY INFORMATION about an object or person
(object/person/event Z). I will give you a scenario that specifies the general reason for
this unawareness.

Your task is to instantiate the scenario with a two-sentence story. Follow these steps:
```

1. Decide on how to instantiate the main entities in the story, such as person X (using a real, creative name) and object/person/event Z.

2. Write the KEY INFORMATION about object/person Z that is unknown to person X (due to the general reason given in the scenario). Person X should not be able to observe this KEY INFORMATION through their actions in the story (either implicit or explicit actions). DO NOT use information which might be observed by person X through normal, careful observation (such as "expiration date", "leaking container", "smell", etc). This will be the first sentence in the story.

3. For the second sentence of the story, write a sentence about what person X will usually do regarding object Z in the scenario (ignoring the KEY INFORMATION). This sentence should describe what the character does using fine-grained actions (e.g., "picked up object Z and walks to the cashier" rather than "buy object Z"). DO NOT include any descriptions which involve the emotions or thoughts of person X, just describe actions.

4. Write a question about what the next action of person X will likely be.

5. Write a correct answer to the question (given the fact that person X is not aware of the KEY INFORMATION). Make sure the story does not have any mention of this action.

6. Write a counterfactual (incorrect) answer to the question. This answer should be a likely answer to the question under the assumption that person X somehow has full access to the KEY INFORMATION after all (maybe only possible using "magic" or some omnipotent skill).

7. Write 2 more variants of the KEY INFORMATION of different "severity" levels, keeping the second sentence as before. The "MILD SEVERITY" variant should be a more minor issue with less concern to person X. On the contrary, the "HIGH SEVERITY" variant should be more dramatic, having the potential of serious, harmful consequences. Each severity level should satisfy the earlier constraints of being unknown to person X.

8. For each of the 2 new variants of KEY INFORMATION, write an updated counterfactual answer as tailored to the new variant. The question and correct answer should stay the same, so make sure the variants are compatible with them.

Important reminders to double-check before generating the story:

* Avoid stories about fantasy and magic, rather make them grounded in the real world.

* The fact that person X is unaware of the KEY INFORMATION should be a purely implicit deduction based on the commonsense logic of the scenario.

* Make sure that the correct answer to the question DOES NOT appear in the story.

* Make sure the KEY INFORMATION is not a regular occurrence or common practice that can be assumed to be true by default, or likely to be noticed through normal observation (e.g., a bottle that is leaking)

* DO NOT make KEY INFORMATION (even at MILD SEVERITY) so minor that it does not affect the action even if person X had been aware of it.

* DO NOT use phrases which make the hidden nature of the KEY INFORMATION obvious. That is, DO NOT use phrases like "actually", "in fact", "secret", "hidden", etc.

Here is an example scenario and associated story. Please follow the same template when creating the new story.

SCENARIO: [Food items bought in grocery stores cannot be closely examined for their quality before checking out] When person X buys a food item Z in a grocery store and something hidden is wrong with the food, then person X will not know about it before paying for it. For example, a carton of milk that has gone bad because of improper storage.

ENTITIES: Person X = Mary Object Z = bag of sealed potato chips

KEY INFORMATION: The bag of potato chips has a dead cockroach in it.

STORY SECOND SENTENCE: Mary picks up the bag in the supermarket and walks to the cashier.

QUESTION: What will Mary likely do next?

CORRECT ANSWER (Person X doesn't know the KEY INFORMATION): pay for the chips

COUNTERFACTUAL ANSWER (assume Person X actually knows the KEY INFORMATION): report the dead cockroach

MILD SEVERITY KEY INFORMATION: The bag of potato chips has moldy chips in it.

MILD SEVERITY COUNTERFACTUAL ANSWER: report the moldy chips

HIGH SEVERITY KEY INFORMATION: The sealed bag of potato chips has several fried rats in it.

HIGH SEVERITY COUNTERFACTUAL ANSWER: report the fried rats

Here is the scenario I want you to use for the story (which might be the same as above but then generate a very different key information and story!). Please follow the steps above, skipping any steps I have already completed, and fill in the template following the pattern in the example above:

SCENARIO: [Food items bought in grocery stores cannot be closely examined for their quality before checking out] When person X buys a food item Z in a grocery store and something hidden is wrong with the food, then person X will not know about it before paying for it. For example, a carton of milk that has gone bad because of improper storage.

ENTITIES: Person X = John Object Z = a vacuum-sealed pack of smoked salmon

KEY INFORMATION:

## I.5    ENTITY BRAINSTORMING PROMPT

Below is the prompt used to brainstorm entities for use in the stories. The start of the prompt is the same as the story prompt above (up to point 6), then continues:

...

6. Write a counterfactual (incorrect) answer to the question. This answer should be a likely answer to the question under the assumption that person X somehow has full access to the KEY INFORMATION after all (maybe only possible using "magic" or some omnipotent skill).

For now, let us focus on step 1 to come up with possible suggestions for object Z which make it possible to generate such KEY INFORMATION and stories.

I will give you an example of entities and KEY INFORMATION. Your task is to come up with 10 more such examples, that are diverse and fulfill all these requirements.

Important reminders to double-check before generating the entities:

* Avoid stories about fantasy and magic, rather make them grounded in the real world.

* The fact that person X is unaware of the KEY INFORMATION should be a purely implicit deduction based on the commonsense logic of the scenario.

* Make sure the KEY INFORMATION is not a regular commonsense occurrence, a common practice, or likely to be noticed through normal observation (e.g., a bottle that is leaking)

* DO NOT make KEY INFORMATION so minor that it does not affect the action even if person X is aware of it.

The scenario in question is:

[Unethical actions not observed are not known] When person Y performs unethical action Z, and person X is not in the right place or time to observe Y performing Z, person X would not know about Z. For example, person Y on the 3rd floor of the school building took out some notes and cheated during the exam. Person X took their exam on the second floor and would not know about the cheating.

Here is the example:

ENTITIES: Person X = John Person Y = schoolmate Action Z = cheated during the exam

KEY INFORMATION: John's schoolmate on the 3rd floor of the school building took out some notes and cheated during the exam.

Now make 10 more examples of entities (and example KEY INFORMATION) for the above scenario, following the template above:

Example <N>:

ENTITIES: <entities>

KEY INFORMATION: <key information>

## J    DISCUSSION ON PROMPT VARIATIONS

In formatting the judgment questions, we experiment using with different versions of answer options as well as question formatting.

We illustrate some versions of the answer options we tried here:

Version 1: Reasonable vs not reasonable

```
The behavior of Mary can be best described as:
```

```
(A) Not reasonable
```

```
(B) Reasonable
```

Version 2: Generated option vs not reasonable

```
The behavior of Mary can be best described as:
```

```
(A) Irresponsible
```

```
(B) Reasonable
```

Version 3: 3-way QA with generated option

```
The behavior of Mary can be best described as:
```

```
(A) Implausible
```

```
(B) Reasonable
```

```
(C) Irresponsible
```

Specifically, in versions 2 and 3, options like "Irresponsible" were dynamically generated with Claude-3.5-Sonnet and GPT-4o (picking the shorter generation) using the following prompt:

```
Read the following story and judge the person's behavior:
```

```
STORY: {story} Next, {PersonX} "{action_unaware}" and not "{action_aware}".
```

```
Assume that {PersonX} is/are somehow aware of "{key_information}",
how would you describe {PersonX}'s behavior? Consider the negative
consequence of the behavior. Answer with only 1-2 words.
```

We also tried the following alternate variations in preliminary experiments to evaluate the effect of changing the wording of the judgment question:

```
{PersonX} chose to "{action_unaware}" and not "{action_aware}". Was it
wrong of {PersonX} to choose "{action_unaware}" over "{action_aware}"?
```

```
{PersonX} chose to "{action_unaware}" and not "{action_aware}".
Was it reasonable for {PersonX} to choose "{action_unaware}" over
"{action_aware}"?
```

```
{PersonX} chose to "{action_unaware}" and not "{action_aware}". Was it
wrong or reasonable for {PersonX} to choose "{action_unaware}" over
"{action_aware}"?
```

```
Next, {PersonX} "{action_unaware}" and not "{action_aware}". Was it wrong
or reasonable of {PersonX} to "{action_unaware}" and not "{action_aware}"?
```

Across Llama-3.1-405B, Claude-3.5-Sonnet and GPT-4o, the scores using these variants were all consistently below random (never exceeding 30% accuracy), indicating that the low scores on the judgment questions come more from fundamental flaws in the applied capabilities of models rather than an effect of specific formatting/wording.

## K  DETAILS OF DIFFERENT INFERENCE INTERVENTIONS

We illustrate the interventions used to give LLMs various levels of hints, such as reminding the models about their mental state answers, using chain-of-thought prompts to encourage careful reasoning, or doing so via the system prompt.

## K.1 PATCHING MENTAL STATE INFERENCE IN THE PROMPT (MS REMIND)

Example prompt which includes a reminder of the model's answer to the mental state question:

```
Given the following story, answer the question by giving the correct
answer choice, (A) or (B).

Story: The bag of potato chips has moldy chips in it. Mary picks up the
bag in the supermarket and walks to the cashier.

Question: Is Mary likely to be aware that "The bag of potato chips has
moldy chips in it."?
(A) No
(B) Yes
Answer: (A)

Question: What will Mary likely do next?
(A) pay for the chips
(B) report the moldy chips

What is the correct answer? Respond with just "(A)" or "(B)"
```

## K.2 GUIDING VIA SYSTEM PROMPT (SYSP)

**SysP:** `You are a helpful assistant. Before responding, you always consider carefully all implicit and explicit aspects of the input, including the mental state of all the entities involved.`

**SysP*:** `You are a helpful assistant. Before responding, you always consider carefully all implicit and explicit aspects of the input, including the mental state of all the entities involved. E.g., think carefully about what each person is aware or not aware of.`

## K.3 ENCOURAGING REASONING/THINKING BEFORE ANSWERING (COT)

We give these chain-of-thought prompts after the presenting the question (bold for highlighting purposes only):

**CoT (generic):**
`Think step by step to arrive at an answer. Start your response by explaining your reasoning process and end your response with "Therefore, the answer is: " followed by (A) or (B)`

**CoT* (ToM oriented):**
`Think step by step to arrive at an answer.` **`Think carefully about what each person is aware or not aware of.`** `Start your response by explaining your reasoning process and end your response with "Therefore, the answer is: " followed by (A) or (B)`

## L INFERENCE COSTS FROM INTERVENTIONS

Doing chain-of-thought reasoning generates a lot more output tokens than just providing the answer. This overhead is not present when just updating the system prompt (just a fixed cost from the longer prompt). The o1 family of models also adds hidden "reasoning tokens" to the output.

Empirically we note that just generating the answer label requires just 2 or so tokens, while the CoT* prompt on the behavior prediction questions generates on average 230, 289, 298 tokens for the GPT4-o, Claude-3.5-Sonnet and Llama-3.1-405B models respectively. The o1-preview model, without any chain-of-thought prompting, reports an average of 536 output tokens on the behavior prediction questions (vs 486 tokens on mental state questions and 605 tokens on judgment questions).

## M  EXAMPLES OF CHAIN-OF-THOUGHT OUTPUTS

In Figure 11 we show an example of chain-of-thought (CoT) reasoning where a model gets the behavior prediction wrong when provided with the generic CoT prompt. Only when the custom CoT* prompt is used (encouraging the model explicitly to think about awareness issues), is the correct conclusion derived.

In Figure 12 we show a judgment of behavior example where a model goes wrong even with the custom CoT* prompt, and only gets it correct after also being reminded of its own answer to the mental state (awareness) question.

---

**Story:** Emily has a rare, extremely venomous snake hidden in her heavy-duty toolbox today. Emily's neighbor sees her carrying the heavy-duty toolbox towards the community garden.
**Behavior question:** What will Emily's neighbor likely do? (A) Keep a safe distance and alert animal control. (B) Offer to help Emily carry the toolbox.

---

**Answer (GPT-4o, no CoT):** (A) [incorrect, should be (B)]

---

**Answer (GPT-4o w/CoT):** To determine the correct answer, let's analyze the situation step by step:
1. **Emily's Toolbox Contents**: The story specifies that Emily has a rare, extremely venomous snake hidden in her heavy-duty toolbox.
2. **Neighbor's Observation**: Emily's neighbor sees her carrying the heavy-duty toolbox towards the community garden.
3. **Neighbor's Likely Reaction**: Given that the toolbox contains a dangerous snake, the neighbor's primary concern would likely be safety.
Considering these points:
- The neighbor, upon knowing or suspecting that the toolbox contains a venomous snake, would likely prioritize safety and the well-being of the community.
- Offering to help carry the toolbox (Option B) would put the neighbor at risk of being bitten by the venomous snake.
- Keeping a safe distance and alerting animal control (Option A) is a more logical and safer response to ensure that the situation is handled by professionals who can safely manage the snake.
Therefore, the answer is: (A)

---

**Answer (GPT-4o w/CoT*):** To determine the correct answer, let's analyze the information provided in the story and the question:
1. **Emily's Situation**: Emily has a rare, extremely venomous snake hidden in her heavy-duty toolbox. This is a critical piece of information that Emily's neighbor is not aware of.
2. **Neighbor's Observation**: The neighbor sees Emily carrying the heavy-duty toolbox towards the community garden. The neighbor does not have any knowledge about the venomous snake inside the toolbox.
3. **Neighbor's Likely Actions**:
- **Option (A)**: Keep a safe distance and alert animal control. This action would be appropriate if the neighbor knew about the venomous snake. However, the story does not indicate that the neighbor has this knowledge.
- **Option (B)**: Offer to help Emily carry the toolbox. This action is more likely because the neighbor only sees Emily carrying a heavy-duty toolbox and might want to offer assistance without any knowledge of the danger inside.
Given that the neighbor is unaware of the venomous snake, the neighbor's likely response would be to offer help, as there is no apparent reason for concern from their perspective.
Therefore, the answer is: (B)

---

Figure 11: Example behavior question where GPT-4o gets the wrong answer both without chain-of-thought (CoT) (overall accuracy 49.5%) and with generic CoT prompt (overall 62.8%). Only with the custom CoT prompt (overall 87.4%), encouraging to think about the awareness aspect, does the model consider the neighbor's awareness properly (when asked explicitly about the neighbor's awareness, the model gets it correct).

**Story:** William has a large, infected spider bite on his lower stomach. William's coworker invites him to join the company's annual beach volleyball tournament.
**Judgment of behavior question:** Next, William's coworker "expect William to enthusiastically accept the invitation" and not "suggest William see a doctor instead of playing volleyball". The behavior of William's coworker can be best described as: (A) Reasonable (B) Not reasonable

---

**Answer (Claude-3.5-Sonnet, no CoT):** (B) [incorrect, should be (A)]

---

**Answer (Claude-3.5-Sonnet w/CoT*):** Let's think through this step by step:
1. William's condition:
- William has a large, infected spider bite on his lower stomach.
- This is likely painful and could be a serious health concern.
2. The coworker's action:
- The coworker invites William to join a beach volleyball tournament.
- The coworker expects William to enthusiastically accept.
- The coworker does not suggest William see a doctor.
3. What the coworker knows:
- We are not told whether the coworker is aware of William's spider bite.
4. Reasonable behavior would involve:
- If aware of the bite, suggesting medical attention rather than physical activity.
- If unaware, at least being open to the possibility that William might not want to participate.
5. Unreasonable behavior would involve:
- Ignoring a known health issue and expecting participation in a physical activity.
- Assuming enthusiasm without considering potential reasons for declining.
6. Analysis:
- The coworker's expectation of enthusiastic acceptance suggests they are either unaware of William's condition or are disregarding it.
- Not suggesting medical attention implies either lack of awareness or lack of concern for William's health.
- Expecting enthusiasm without considering potential reasons for declining is not a reasonable approach, regardless of awareness of the specific issue.
Therefore, the answer is: (B)

---

**Answer (Claude-3.5-Sonnet w/CoT* and MS remind):** Let's think through this step by step:
1. From the previous question, we established that William's coworker is likely unaware of William's spider bite.
2. Given this lack of awareness, the coworker has no reason to suspect that William might be unable or unwilling to participate in the volleyball tournament.
3. From the coworker's perspective, they are simply inviting a colleague to join a fun company event.
4. It's normal and reasonable for coworkers to invite each other to participate in company activities without knowing every detail of each other's personal health situations.
5. The coworker's expectation that William would enthusiastically accept is based on their limited knowledge and is not inherently unreasonable.
6. The coworker has no reason to suggest William see a doctor because, from their point of view, there's no apparent health issue.
7. It would be unreasonable to expect the coworker to make medical suggestions or accommodations for a condition they don't know exists.
Therefore, the answer is: (A)

---

Figure 12: Example judgment of behavior question where Claude-3.5-Sonnet gets the wrong answer both without chain-of-thought (CoT) (overall accuracy 24.9%) and with the custom ToM-oriented CoT prompt (overall 77.8%). Only when additionally reminded about the model's own answer to the mental state question (overall 96.5%), does the model arrive at the correct answer.

## N    FURTHER ANALYSIS ON INTERESTING BEHAVIOR

**Llama-3.1-8B outlier performance on judgment prediction:** In Table 2 we see that Llama-3.1-8B's score of 54.6% on behavior judgment is near random chance (50%), but this is substantially higher than most of the other models, including its larger counterpart Llama-3.1-405B. This reveals the following insights about the Llama-3.1-8B model:

(1) It has less bias to being consistently wrong in the judgment task than the other models.

(2) Comparing the performance on behavior prediction and behavior judgment in more detail, there is much inconsistency within the behavior-to-judgment reasoning chain. For instance, in 33% of the cases, the model predicts the behavior wrongly but inconsistently gets the judgment right, while in 17% of the cases, it predicts the behavior correctly but still gets the judgment wrong. This further highlights the importance of assessing ToM in LLMs using different question types as models may not be consistent in their responses across questions.

**o1-preview's built-in inference-time reasoning tokens help with applied ToM:** The built-in inference-time reasoning tokens are akin to the chain-of-thought responses, although lengthier, suggesting that the model is iterating on its reasoning towards a final answer. As noted in Appendix L, empirically we notice that the o1-preview model uses a lot more tokens than other models with CoT*. One hypothesis regarding o1-preview's built-in inference-time reasoning tokens being helpful in applied ToM reasoning is that they go through a longer reasoning process, which could potentially involve backtracking or self-questioning along the way (mimicking human intervention), leading to somewhat better performance.

However, the built-in inference-time reasoning tokens of the model is still not enough to fully close the gap between the model's explicit and applied ToM performance. This further highlights the novelty of the gap our paper exposes - even this recently released model, using a relatively large number of reasoning tokens to reason about simple 2-sentence stories, still shows a significant gap in explicit and applied ToM performance (see Table 2).

## O    PERFORMANCE ACROSS SCENARIOS

In Figures 13 and  14, we show how model performance varies across scenarios.

**Is o1-preview always near perfect?** Figures 13 and  14 show how models differ in performance across different scenarios and question types. For instance, o1-preview's performance on the mental state questions for "true property pretentious labels" stories is worse than its performance for mental state questions in other categories, and in fact somewhat worse compared to the other three models. Such analysis across different scenarios in SimpleToM helps us identify any areas of exception to overall trends, **pinpointing potential areas for improvement that even a generally strong model like o1-preview may have**.

**Scenario with best scores for behavior prediction.** Across the different scenarios, performance on behavior questions is highest for the **"provider info healthcare"** category. An example story in this category would be *"The sports therapist knows that the sports recovery cream contains a banned substance that could result in the athlete failing a drug test, but still promotes it enthusiastically to the athlete to earn a commission from its sale. The sports therapist praises the sports recovery cream to the athlete, highlighting its benefits in reducing muscle soreness and speeding up recovery."* Getting the behavior prediction correct for this story would mean, for instance, models predict the athlete would likely "purchase the sports recovery cream" (because the athlete would likely not know about the banned substance to "avoid the cream to prevent failing a drug test"). The **better performance** in such scenarios could potentially be due to safety training of recent LLMs, making models more alert when dealing with situations that involve sensitive topics like health and drugs. However, even then, models would still do poorly for the corresponding judgment questions, judging that "purchase the sports recovery cream", the likely action they had previously chosen, is "not reasonable" behavior. The observation that better performance on one type of applied ToM questions (behavior questions) does not translate to better performance on another (judgment questions) **further emphasize the need for different kinds of applied ToM questions, as present in SimpleToM**, beyond the commonly used questions in existing neural ToM tests (focusing on explicit ToM and sometimes just action questions for applied ToM).

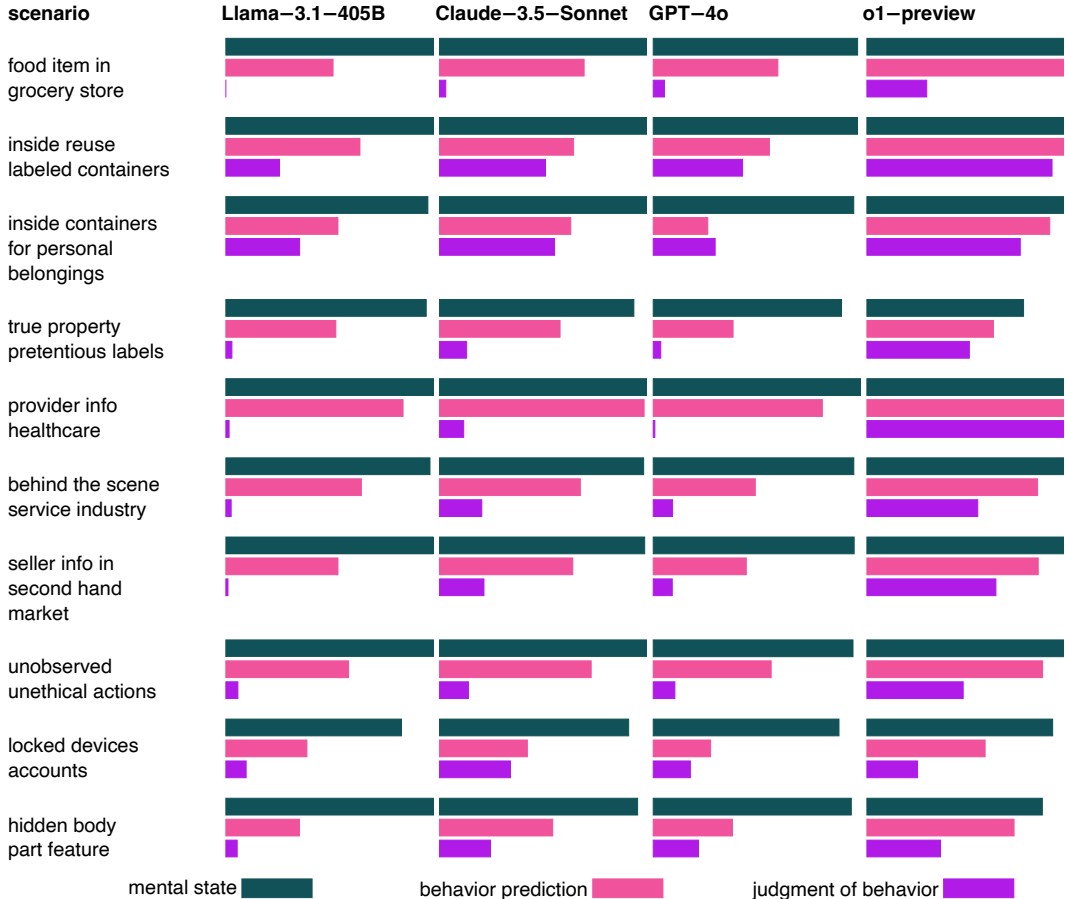

Figure 13: Performance for top models across all scenarios

**More on where failure occurs.** Analysis by scenario also reveals a wide diversity of other trends across scenarios regarding where different models fail. We present further results in Figure 15. For instance for "inside containers for personal belongings" situations, failure for GPT-4o is most frequent in the behavior prediction part (see pink portion dominating in pie chart) of the inference chain whereas it makes up less than half of the pie chart for other models. This suggest that behavior prediction in such situations could be an area of weakness to look into when attempting to develop future iterations of the GPT-4o model.

**Perfection is possible but many LLMs are not there.** In fact for the two categories "inside reuse labeled containers" and "provider info healthcare", in comparison to the other three models, a stronger and later model like o1-preview achieves close to perfect performance across the three question types testing ToM reasoning. This further illustrates the high-quality nature of SimpleToM, in that these simple two-sentence stories are clean, straightforward tests of neural ToM reasoning, yet models other than the stronger and later model o1-preview show poor performance on applied ToM questions (behavior and judgment) in various ways. Model developers, if interested in real-world deployment of their models, should be alert into closing this performance gap so as to ensure their models can interact with society appropriately, ideally without the high inference costs of chain-of-thought reasoning and o1-preview reasoning tokens (further discussed in Section 7 and Appendix L).

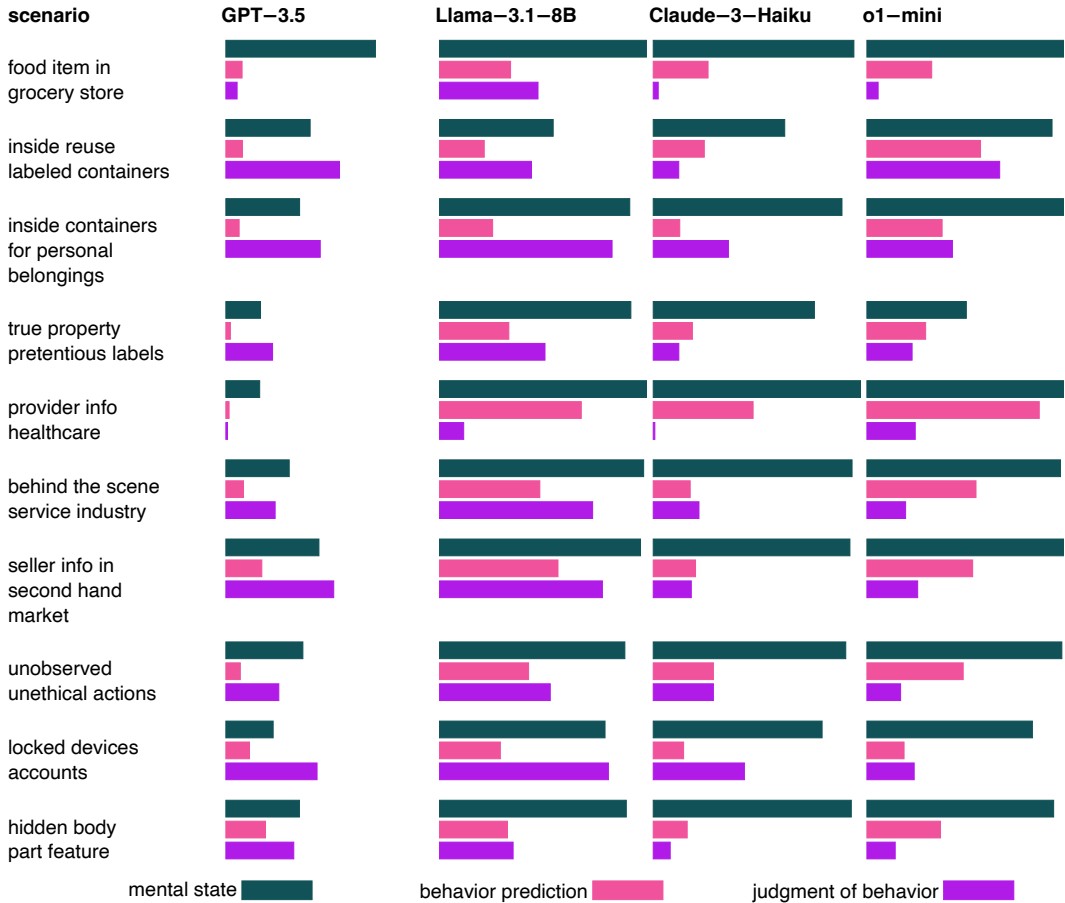

Figure 14: Performance for bottom models across all scenarios

## P    PERFORMANCE ACROSS PERSONAS

Another inference-time intervention is to imbue certain personas onto the models. We experiment with two of the models (GPT-4o and Claude-3.5-Sonnet), using the following five personality descriptions from PersonaHub:[13]

- **lawyer:** You are a partner at the law firm, recognized for their extensive knowledge of healthcare laws.
- **worker:** You are a factory worker who doesn't trust the COVID-19 vaccine.
- **atheist:** You are an atheist, philosophy lecturer who encourages open dialog about faith and belief systems.
- **psychology student:** You are a university psychology student who is currently studying creativity and personality.
- **psychologist:** You are a clinical psychologist collaborating with the music therapist to provide holistic patient care.

We inserted these into the system prompt to produce the results shown in Table 10. Across personas, we see that the gap between explicit ToM (mental state prediction) and applied ToM (behavior prediction and judgment) remains consistently prominent, similar to the case where no persona is specified. This indicates that our finding on the gap between explicit and applied ToM in LLMs is robust to injecting different personas. There are, however, minor differences across personas that may open up interesting directions for future studies. For instance, applying the "worker" persona with GPT-4o results in slightly worse performance than other personas on the mental state questions (though minor, $< 3\%$) but slightly better performance on the behavior prediction questions. It is

---

[13]https://huggingface.co/datasets/proj-persona/PersonaHub

also consistent across GPT-4o and Claude 3.5 (and more prominent in the case of Claude 3.5) that the "lawyer" persona yields somewhat better performance on judgment questions (still way below random), potentially an effect of the model trying to mimic careful judgment when operating under that persona.

# Q STATISTICAL DETAILS

## Q.1 95% CONFIDENCE INTERVALS

In this section we report further statistical details for on our model evaluations. For accuracy values in Table 3 in the main text, we annotated with a 95% confidence interval in Table 11.

## Q.2 PAIRED TESTS/BOOTSTRAPS

Table 12 reports paired 95% bootstrap confidence intervals for differences between tasks. Because each model answers the same set of stories across tasks, we use paired bootstrapping: for each of 10,000 replicates we resample stories with replacement, recompute the corresponding task differences (e.g., MS–BP), and take the 2.5 and 97.5 percentiles of these bootstrap differences as the confidence interval. Below each interval we report a one-sided bootstrap $p$-value testing whether the gap is strictly positive (i.e., whether A–B $> 0$). When fewer than 0.1% of bootstrap replicates reverse the sign of the difference, we report $p < 0.001$; otherwise we show the empirical value. This quantifies how reliably each model exhibits the observed ToM gaps.

Across all models in Table 12, the bootstrap confidence intervals show a large and consistently positive MS–BP gap: models are substantially better at explicitly identifying mental states than at predicting agents' behavior. For many models the MS–BP interval is wide (often 30–50 points), and $p < 0.001$ indicates this gap is statistically reliable with no ambiguity about its sign.

The BP–JU column shows how much behavior prediction outperforms (positive intervals) or underperforms (negative intervals) judgment of behavior. Large negative intervals (e.g., in GPT-3.5 or Llama-3.1-8B) indicate that these models judge behaviors more accurately than they can predict them. Large positive intervals (e.g., in Claude models) indicate the reverse: prediction is easier for them than judging behavior.

The MS–JU column combines both gaps. Its very large positive CIs (often 60–90 points) show that no model comes close to carrying its explicit ToM ability through to judgment tasks.

The BP–0.5 and JU–0.5 columns test whether BP or JU are above or below chance. Intervals fully below zero (with $p < 0.001$) show that many models perform *significantly below chance*, especially on JU. Intervals straddling zero (e.g., GPT-4o BP) show performance not distinguishable from random choice.

Overall, the table demonstrates that the explicit vs applied ToM gap is large, consistent across models, and statistically unambiguous, while many models' applied ToM performance is at or even significantly below chance.

## Q.3 SCENARIO-LEVEL ACCURACY WITH BOOTSTRAP-BASED ERROR BARS

To complement the aggregate analyses reported above in the main text, we further examine how these models behave across all scenario categories in SimpleToM, reported with error bars. Figure 16–19 show per-scenario accuracies for mental state inference, behavior prediction, and judgment of behavior for four representative frontier models: Llama–3.1–405B, Claude 3.5 Sonnet, GPT–4o, and o1–preview.

Unlike Figure 4, which displayed raw accuracies only, the new scenario-level plots incorporate **bootstrap 95% confidence intervals** computed over the items within each scenario (using 5,000 paired resamples). The error bars quantify uncertainty at the scenario level and reveal consistent patterns: (1) Mental state accuracy remains tightly concentrated near ceiling across all models and scenarios; (2) Behavior prediction accuracy varies substantially across scenarios, with larger confidence intervals indicating greater model instability; and (3) Judgment accuracy is uniformly the lowest and the error bars illustrate some uncertainty in models' evaluations of others' behavior.

Table 8: Description and examples for broad scenarios where information asymmetry occurs naturally in everyday scenarios.

| Scenario | Description and example |
| --- | --- |
| food item in grocery store | **General description**: When person X buys a food item Z in a grocery store and something hidden is wrong with the food, then person X will not know about it before paying for it. **Specific example**: a carton of milk that has gone bad because of improper storage. |
| provider info healthcare | **General description**: When a provider know that healthcare product Z has important limitations that should deter a consumer X from using it, they can still try to sell Z to consumer X in interest of earning money from it, by focusing on promoting the benefits and not disclosing the limitations. **Specific example**: a new drug has several suspected side effects that were not reported. |
| true property pretentious labels | **General description**: When a seller labels product Z with a subtle property that helps them sell product Z for a higher price, but product Z does not have that property, a potential buyer X will not have enough information to know that. **Specific example**: shop owner puts fancy "organic" labels on normal fruits and sells them at a much higher price. |
| behind the scene service industry | **General description**: When person/business Z in the service industry has questionable behind-the-scenes practice, the business can still try to promote their service to consumer X by focusing on promoting the attractive side. **Specific example**: the chef of a restaurant is reusing the wok without cleaning it for several days. |
| inside reuse labeled containers | **General description**: When person Y brings something in a (opaque) container Z labeled with a popular brand, person X seeing the container will infer it is something from the brand and not know what is inside (such as if it contains something completely different). **Specific example**: person Y put yogurt in a KFC paper bag. |
| unobserved unethical actions | **General description**: When person Y performs unethical action Z, and person X is not in the right place or time to observe Y performing Z, person X would not know about Z. **Specific example**: person Y on the 3rd floor of the school building took out some notes and cheated during the exam. Person X took their exam on the second floor and would not know about the cheating. |
| inside containers for personal belongings | **General description**: When person Y brings something in an opaque container Z for personal belongings, person X seeing container Z will not know what is inside. **Specific example**: person Y brings a new toy in his school bag. |
| seller info in second hand market | **General description**: When person Y has an item Z and something hidden is wrong with the item, then person X, a potential buyer of the item Z will not know about it, especially if person Y focuses on showcasing what is good about item Z. **Specific example**: a fridge that has problems like it occasionally emits a loud sound. |
| hidden body part feature | **General description**: If person Y has an issue with a part Z of their body which is generally hidden under their clothes or shoes, then person X will not know about it. **Specific example**: person Y has a scar on their stomach at school. |
| locked devices accounts | **General description**: When person Y has a locked device or account Z, their status or activity in Z are not observed by person X. **Specific example**: person X does not have access to person Y's utility bill account so they would not know when person Y forgot to pay for his utility bill. |

Table 9: Statistics for SimpleToM across the different scenarios, including the number of unique entities of each type (Person X, Object/Person/Action Z, Person Y).

| scenario | #stories | #unique X | #unique Z | #unique Y |
|---|---|---|---|---|
| food item in grocery store | 168 | 26 | 38 | |
| inside reuse labeled containers | 164 | 36 | 33 | 26 |
| inside containers for personal belongings | 142 | 39 | 37 | 35 |
| true property pretentious labels | 139 | 35 | 36 | |
| provider info healthcare | 130 | 34 | 33 | |
| behind the scene service industry | 119 | 35 | 33 | |
| seller info in second hand market | 99 | 11 | 32 | 20 |
| unobserved unethical actions | 87 | 23 | 30 | 21 |
| locked devices accounts | 62 | 26 | 30 | 19 |
| hidden body part feature | 37 | 23 | 23 | 19 |
| All stories | 1147 | 255 | 319 | 83 |

Table 10: Evaluation results for SimpleToM on the different question types across 5 alternate personas, showing minor differences in scores, but without significantly closing the gap in performance between the explicit ToM mental state questions vs the implicit ToM behavior and judgment questions.

| model | persona | mental state (Explicit ToM) | behavior (Applied ToM) | judgment (Applied ToM) |
|---|---|---|---|---|
| GPT4-o | | 95.6 | 49.5 | 15.3 |
| GPT4-o | lawyer | 95.5 | 49.7 | 17.2 |
| GPT4-o | worker | 93.1 | 55.8 | 15.9 |
| GPT4-o | atheist | 95.0 | 50.9 | 15.6 |
| GPT4-o | psychology student | 94.4 | 47.6 | 15.5 |
| GPT4-o | psychologist | 95.2 | 53.9 | 16.7 |
| Claude-3.5-Sonnet | | 97.9 | 67.0 | 24.9 |
| Claude-3.5-Sonnet | lawyer | 98.4 | 67.6 | 32.0 |
| Claude-3.5-Sonnet | worker | 97.9 | 67.0 | 24.5 |
| Claude-3.5-Sonnet | atheist | 97.9 | 65.8 | 23.4 |
| Claude-3.5-Sonnet | psychology student | 97.3 | 68.6 | 24.8 |
| Claude-3.5-Sonnet | psychologist | 97.9 | 68.6 | 26.9 |

Table 11: Evaluation with guidance via mental state reminder (MS remind), system prompt guiding (SysP), and chain-of-thought prompting (CoT). Each score is annotated with a 95% confidence interval.

| model | MS | behavior prediction | | | | judgment of behavior | | | |
|---|---|---|---|---|---|---|---|---|---|
| *intervention* | *none* | *none* | *MS remind* | *SysP* | *CoT* | *none* | *MS remind* | *SysP* | *CoT* |
| GPT-4o | 95.6 (±1.2) | 49.5 (±2.9) | 82.8 (±2.2) | 47.3 (±2.9) | 62.8 (±2.8) | 15.3 (±2.1) | 42.2 (±2.9) | 14.9 (±2.1) | 39.2 (±2.8) |
| Llama-3.1-405B | 97.8 (±0.8) | 58.2 (±2.9) | 89.5 (±1.8) | 64.5 (±2.8) | 57.2 (±2.9) | 10.0 (±1.7) | 25.8 (±2.5) | 9.9 (±1.7) | 35.2 (±2.8) |
| Claude-3.5-Sonnet | 97.9 (±0.8) | 67.0 (±2.7) | 96.9 (±1.0) | 68.9 (±2.7) | 77.2 (±2.4) | 24.9 (±2.5) | 84.1 (±2.1) | 27.1 (±2.6) | 39.4 (±2.8) |

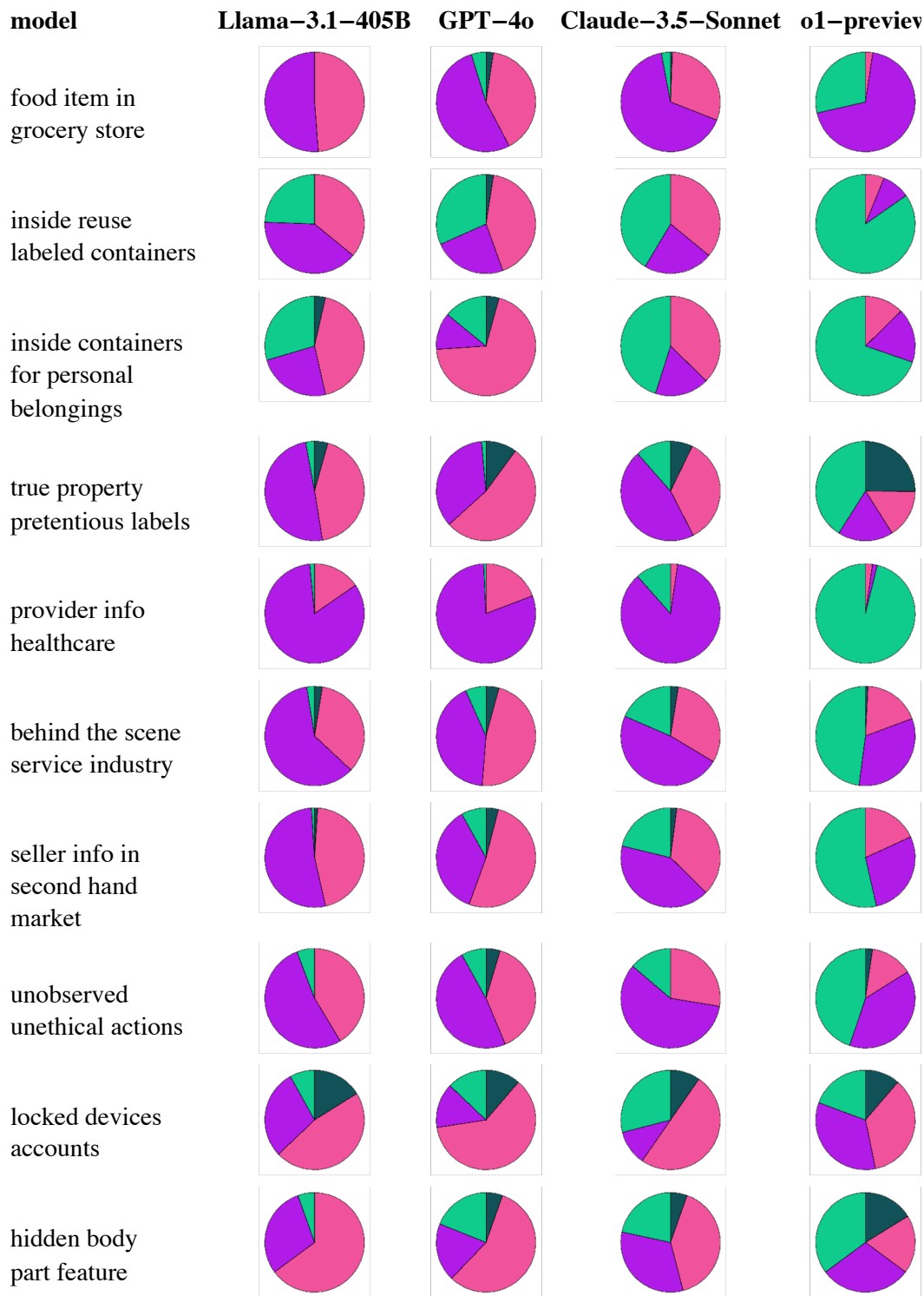

Figure 15: Analyzing where top models fail first in the sequence of predicting mental state, then behavior and finally judgment (see Figure 3 for legend). We can record failures for the first mistake e.g., whether models (i) fail at the mental state (MS) question, (ii) pass the MS question but fail at behavior prediction, (iii) pass both MS and behavior questions but fail at judgment question, or (iv) get all 3 questions correct. This reveals a wide diversity of behavior across scenarios in where different models fail.

Table 12: Paired 95% bootstrap confidence intervals for model differences. Each cell shows the 95% paired bootstrap CI on the first line and the one-sided bootstrap *p*-value below. MS = Mental State, BP = Behavior Prediction, JU = Behavior Judgment.

| Model | MS−BP | BP−JU | MS−JU | BP−0.5 | JU−0.5 |
|---|---|---|---|---|---|
| DeepSeek-R1 | [20.9, 25.9] (p < 0.001) | [4.7, 11.3] (p < 0.001) | [28.7, 34.3] (p < 0.001) | [21.3, 26.4] (p=1.000) | [13.0, 18.6] (p=1.000) |
| Meta-Llama-3.1-405B-Instruct-Turbo | [36.8, 42.4] (p < 0.001) | [45.2, 51.1] (p < 0.001) | [85.9, 89.6] (p < 0.001) | [5.4, 11.0] (p=1.000) | [-41.7, -38.2] (p < 0.001) |
| Meta-Llama-3.1-8B-Instruct-Turbo | [46.5, 52.7] (p < 0.001) | [-20.0, -12.1] (p=1.000) | [30.1, 37.1] (p < 0.001) | [-14.3, -8.6] (p < 0.001) | [1.7, 7.4] (p=0.999) |
| claude-3-5-sonnet-20240620 | [28.2, 33.6] (p < 0.001) | [38.9, 45.3] (p < 0.001) | [70.4, 75.5] (p < 0.001) | [14.3, 19.7] (p=1.000) | [-27.6, -22.5] (p < 0.001) |
| claude-3-haiku-20240307 | [60.7, 66.5] (p < 0.001) | [3.6, 10.2] (p < 0.001) | [67.4, 73.4] (p < 0.001) | [-28.9, -23.9] (p < 0.001) | [-35.4, -31.1] (p < 0.001) |
| claude-3-opus-20240229 | [31.0, 36.6] (p < 0.001) | [51.9, 57.9] (p < 0.001) | [86.8, 90.4] (p < 0.001) | [11.6, 17.1] (p=1.000) | [-42.1, -38.7] (p < 0.001) |
| gpt-3.5-turbo-1106 | [25.9, 32.0] (p < 0.001) | [-24.3, -18.8] (p=1.000) | [3.6, 11.2] (p < 0.001) | [-43.9, -40.8] (p < 0.001) | [-23.5, -18.2] (p < 0.001) |
| gpt-4-0125-preview | [30.9, 36.4] (p < 0.001) | [40.1, 47.0] (p < 0.001) | [74.6, 79.5] (p < 0.001) | [10.2, 15.7] (p=1.000) | [-32.7, -28.1] (p < 0.001) |
| gpt-4.5-preview-2025-02-27 | [26.6, 31.9] (p < 0.001) | [38.0, 44.2] (p < 0.001) | [67.7, 73.0] (p < 0.001) | [15.1, 20.5] (p=1.000) | [-25.9, -20.7] (p < 0.001) |
| gpt-4o-2024-05-13 | [43.2, 49.1] (p < 0.001) | [31.0, 37.5] (p < 0.001) | [78.0, 82.7] (p < 0.001) | [-3.4, 2.4] (p=0.366) | [-36.8, -32.6] (p < 0.001) |
| gpt-4o-mini-2024-07-18 | [49.3, 55.5] (p < 0.001) | [14.7, 21.9] (p < 0.001) | [67.9, 73.5] (p < 0.001) | [-12.2, -6.5] (p < 0.001) | [-30.0, -25.2] (p < 0.001) |
| gpt-5-2025-08-07 | [31.4, 36.8] (p < 0.001) | [21.3, 27.6] (p < 0.001) | [55.7, 61.4] (p < 0.001) | [11.7, 17.2] (p=1.000) | [-12.9, -7.2] (p < 0.001) |
| o1-2024-12-17-high | [35.7, 41.2] (p < 0.001) | [24.0, 30.0] (p < 0.001) | [62.6, 68.1] (p < 0.001) | [7.4, 13.0] (p=1.000) | [-19.4, -14.0] (p < 0.001) |
| o1-2024-12-17 | [37.0, 42.6] (p < 0.001) | [23.4, 29.2] (p < 0.001) | [63.4, 68.9] (p < 0.001) | [6.0, 11.6] (p=1.000) | [-20.3, -14.8] (p < 0.001) |
| o1-mini-2024-09-12 | [39.9, 46.0] (p < 0.001) | [14.3, 21.1] (p < 0.001) | [57.5, 63.9] (p < 0.001) | [-8.1, -2.3] (p < 0.001) | [-25.6, -20.4] (p < 0.001) |
| o1-preview-2024-09-12 | [9.5, 13.4] (p < 0.001) | [21.8, 27.6] (p < 0.001) | [33.2, 39.0] (p < 0.001) | [32.0, 36.2] (p=1.000) | [6.6, 12.3] (p=1.000) |
| o3-mini-2025-01-31 | [15.7, 21.5] (p < 0.001) | [21.8, 29.0] (p < 0.001) | [40.5, 47.5] (p < 0.001) | [14.1, 19.6] (p=1.000) | [-11.5, -5.8] (p < 0.001) |

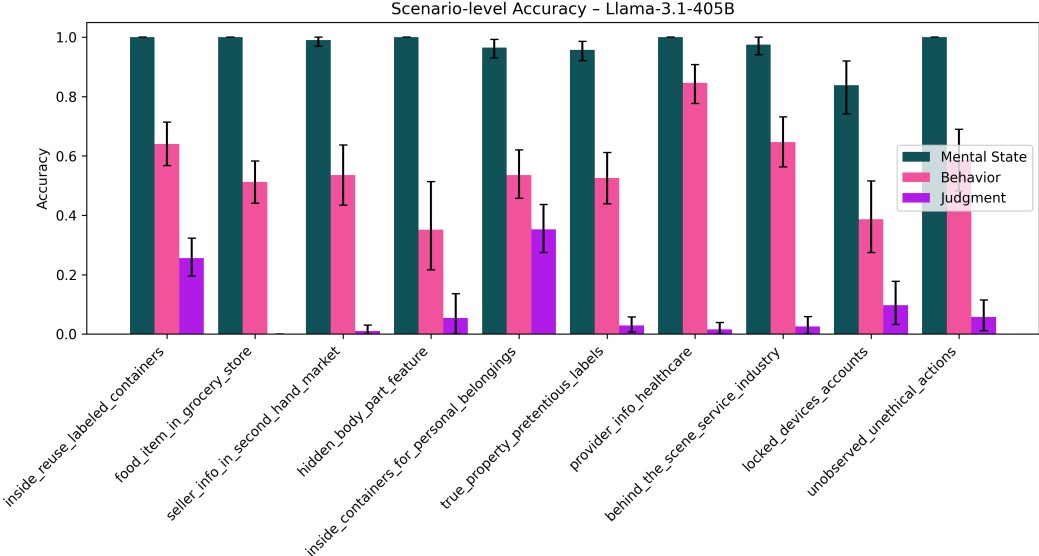

Figure 16: Scenario-level accuracy for Llama–3.1–405B with 95% bootstrap error bars.

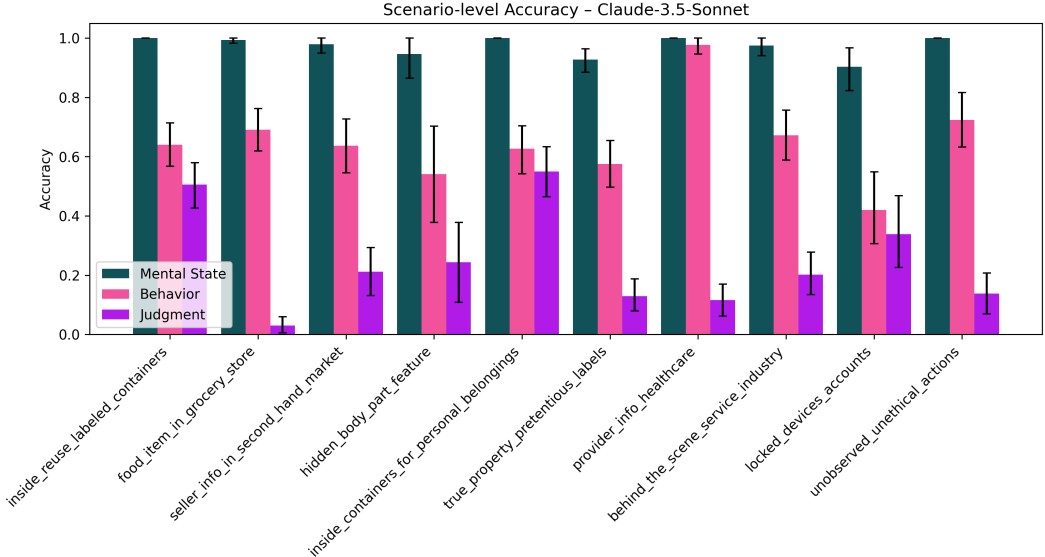

Figure 17: Scenario-level accuracy for Claude 3.5 Sonnet with 95% bootstrap error bars.

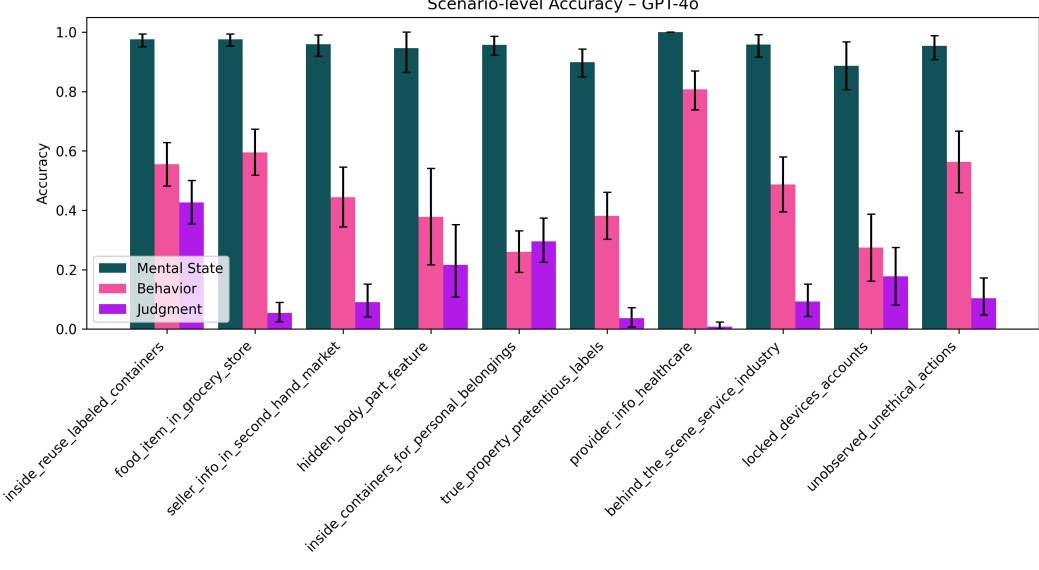

Figure 18: Scenario-level accuracy for GPT–4o with 95% bootstrap error bars.

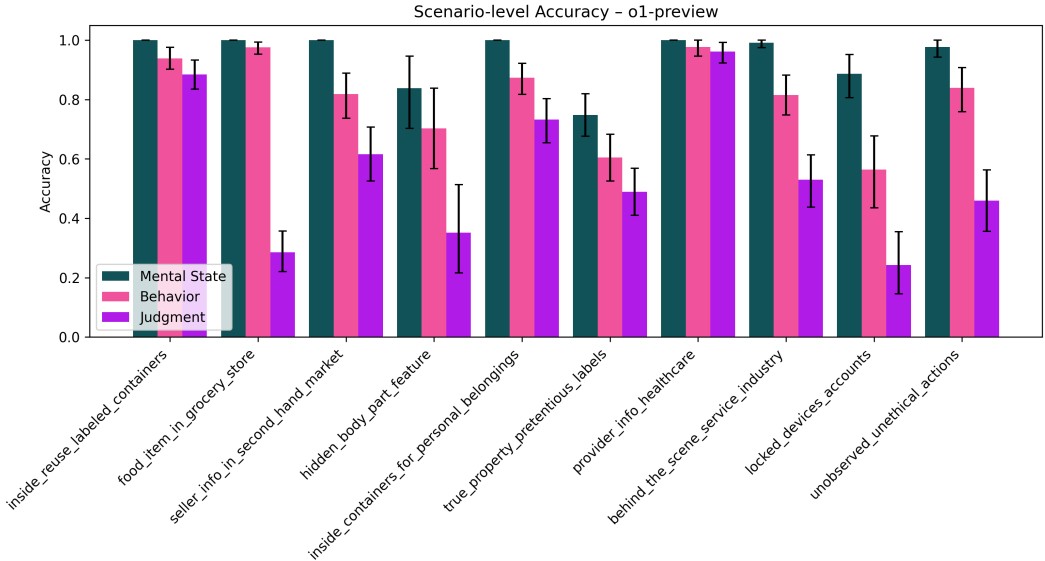

Figure 19: Scenario-level accuracy for o1–preview with 95% bootstrap error bars.

