# OpenReview forum: "SimpleToM: Exposing the Gap between Explicit ToM Inference and Implicit ToM Application in LLMs"
_ICLR.cc/2026/Conference — ICLR 2026 Poster_

### Official Review · Reviewer_6erB · 2025-10-26

**Soundness:** 3
**Presentation:** 3
**Contribution:** 1
**Rating:** 4
**Confidence:** 4

**Summary:**

This work introduces SimpleToM, a benchmark for testing whether LLMs can not only infer mental states (explicit ToM) but also use that understanding to predict and judge behavior (applied ToM).

Results show that (1) while models perform well in inferring mental states, their ability drops sharply for behavior prediction,
(2) model performances across different scenarios differ, and
(3) prompting-based interventions do not fully close the performance gap between implicit and explicit ToM.

The presentation is clear, but my main concern is the originality and significance of the work. I will detail these below.

**Strengths:**

- The presentation is clear, and the experiments are thorough.
- The main distinction between SimpleToM and existing benchmarks appears to be that, based on the example figure and Section 6, SimpleToM does not explicitly describe the characters’ observations. Instead, models must rely on implicit commonsense knowledge. For instance, understanding that “Mary cannot open the chips before paying for them.” This setup is relatively novel and could be valuable for assessing ToM in more natural, less-structured everyday contexts.

**Weaknesses:**

The main concern is that the originality of this work is limited. To my knowledge, BigToM [1] also adopts real world scenarios and includes an “applied ToM” task (the forward action task), suggesting some overlap in contribution.

It is also unclear why models perform poorly on the behavior judgment task even when mental states are given. A possible issue I notice is ambiguous phrasing in the dataset. For example, the question “Next, Mary pays for the chips and not reports the moldy chips. The behavior of Mary can be best described as: A. reasonable, B. not reasonable.” does not clarify who the behavior is reasonable to: Mary herself, or the external observer (we know that there are moldy chips). Such ambiguity may confuse models and affect their judgment.

The authors could examine whether such ambiguity affects performance and consider clearer or more balanced phrasing.

[1] Gandhi, Kanishk, et al. "Understanding social reasoning in language models with language models." Advances in Neural Information Processing Systems 36 (2023): 13518-13529.

**Questions:**

Could you compare the behavior prediction task with the forward action task in BigToM more clearly, since you claim 'applied ToM' is the main contribution of your benchmark?

Could you also explain why the behavior judgment task is needed and what it adds? How is it different from behavior prediction? Is it a new task you designed, or is it based on existing work in psychology / cognitive science?

---

> ### Author Response · Authors · 2025-11-21
>
> We thank the reviewer for their positive feedback on the soundness, clarity, and thoroughness of our work. We’re also glad that you find the distinction between SimpleToM and existing benchmarks—where models must rely on implicit commonsense knowledge—relatively novel and potentially valuable for assessing ToM in more natural, less-structured everyday contexts.
>
> **Originality**
>
> Our main contribution is not just introducing an “applied ToM” task, but designing a new evaluation dataset enabling the community to rigorously access ToM in LLMs and reveal failures that existing benchmarks cannot (See Section 2):
>
> - SimpleToM removes explicit mentalizing cues like BigToM’s “X notices / does not notice”. In BigToM, models can often succeed via keyword matching (“notices” vs “does not notice”). In SimpleToM, stories require commonsense inferences about what characters can perceive or know, closer to how ToM is applied in real life.
>
> - We broaden the coverage of information asymmetry beyond BigToM’s single “witnessed vs unwitnessed event” template. Guided by social psychology (Doherty, 2008), SimpleToM covers different information asymmetries such as manipulation, deception, secrecy, lying, and misleading behavior, including real-world contexts like sales of “lemon” products where hidden flaws are purchased due to lack of information (Akerlof, 1978). Our results show that model performance varies substantially across these scenarios, which cannot be seen from a single template.
>
> - SimpleToM enforces higher realism and consistency: our 3/3 human filtering removes contradictions that appear in BigToM (e.g., “Cheng does not notice the power outage” when he “use[s] a projector to show a documentary”).
>
> - We adopt a three-stage diagnostic design (mental state → behavior → judgment), rigorously guided by psychological frameworks (see Appendix C and D), including Quesque & Rossetti (2020)’s criteria for validating ToM (Chen et al., 2024). This structure aligns with broader cognitive theories contrasting conceptual understanding with procedural/behavioral competence (e.g., Hiebert & Lefevre, 1986; Chomsky, 1959). Following this literature, we intentionally use a simpler two-sentence setup that actively avoids confounders while preserving systematic and scalable data creation.
>
> These choices are what allow us to uncover: (1) the jarring gap between explicit and implicit ToM, (2) large differences across types of information asymmetry, and (3) the failure of prompting-based interventions to fully close these gaps. These findings would not emerge from using existing datasets without the effort that goes into curating SimpleToM.
>
> We agree that BigToM is an important contribution towards allowing studies of ToM to be carried out at scale. We also provide a detailed comparison with BigToM and as well as other datasets (Section 6 related Work version 1 Lines 451 - 469, FAQ “How is SimpleToM different from existing datasets?”).
>
> **Poor model performance on judgment is NOT due to prompting/wording**
>
> In Appendix M and Figure 11, we show a behavior judgment example where a model only succeeds after being explicitly reminded of its own answer to the mental state (awareness) question AND custom ToM-specific CoT* prompt . Through our analysis of models’ CoT output,  models perform poorly on the behavior judgment task even when mental states are given because they do not know to apply the information in the context unless explicitly told to do so using a Theory of Mind task-specific CoT* prompt (general CoT prompt does not suffice), revealing a key fragility of current LLMs in applying information. (We provide more details on such interventions in Appendix F.)
>
> We do not observe this jarring gap in human annotators using the exact same phrasing, therefore this is a limitation due to model capability not wording.
>
> Further, in FAQ and Appendix J, we discuss results from experimenting with different prompt variations for the judgment question, showing that low scores on the judgment questions come from fundamental flaws in the applied ToM capabilities of models rather than an effect of specific
> formatting/wording.

---

> > ### Author Response · Authors · 2025-11-21
> >
> > **Behavior prediction vs BigToM’s forward action**
> >
> > We agree BigToM’s forward action is related in terms of question type, but our behavior prediction task is different as it is embedded in a richer diagnostic graph (Figure 1) where models must first implicitly infer mental states (without explicit cues) and then apply them to predict behavior across diverse, psychologically motivated asymmetries. This is a more rigorous (type of inference required) and holistic (diversity of asymmetries) regime than acting on explicit “notices/does not notice” cues within a single “witnessed vs. unwitnessed event” template. Also see response above for how the construction of stories differ.
> >
> > **Why behavior judgment is needed and what it adds**
> >
> > Behavior judgment involves an additional hop in application of ToM than behavior prediction. As the inference graph in Figure 1 shows, the judgment question goes beyond behavior prediction: it requires (1) implicitly understanding a character’s mental state, (2) implicitly predicting their behavior, and then (3) evaluating whether that behavior is reasonable given their beliefs (See version 1 Lines 222–229). Our goal is to study how models perform when one additional level of reasoning is required, and we find they are extremely fragile even with this single extra step.
> >
> > This design is directly inspired by psychology, where behavior judgment is prevalently used to study ToM, starting from looking at eye gaze patterns in young children to via direct questions for older subjects (Doherty, 2008). In commonsense psychology, people’s mental states are crucial for appropriate moral and social judgments (Jara-Ettinger et al., 2016; Schein & Gray, 2018). SimpleToM explicitly brings these well-established distinctions in stages and components of ToM in psychology literature into LLM evaluation (see Section 6, Appendix C and D for more discussion of this literature), addressing a gap where such careful distinctions are rarely carried over.
> >
> > **Summary**
> >
> > Thank you for your thoughtful review. We hope our clarifications show that SimpleToM’s expanded coverage of information asymmetries, implicit mentalizing design, higher realism via human verification procedure, and psychology-guided three-stage diagnostic structure together introduce new capabilities beyond existing datasets, and are precisely what enable a more rigorous, holistic evaluation that reveals previously unseen weaknesses in LLM social reasoning.

---

### Official Review · Reviewer_xnHV · 2025-10-28

**Soundness:** 3
**Presentation:** 3
**Contribution:** 3
**Rating:** 4
**Confidence:** 4

**Summary:**

This paper investigates a gap in the existing Theory of Mind (ToM) dataset, that most of them focus on the mental state inference while few address the issue of applying the knowledge about the mental state into actions. Inspired by this, this paper constructs a dataset to test the propriety LLMs on how they can infer the mental states as well as how they can apply such inference / knowledge into their actions. The experimental results reveal that behavior predictions present a significant performance drop from mental state inference's prediction. In addition, simple test-time intervention does not seem to help the behavior predictions.

**Strengths:**

- The dataset addresses the gap in ToM research, as few existing datasets investigate the action in ToM space. This paper can serve as the first study in ToM on both the mental state inference and the behavior side study.

- The examples in the dataset are short and easy to evaluate. This can disentangle confounding factors such as the length, verbosity, etc, in ToM evaluation.

- The authors have conducted experiments on proprietary LLMs as well as some open-source LLMs. The results mostly justify their findings.

**Weaknesses:**

- It would be nice for the authors to include more open-source LLMs. Currently, the table 2 includes many proprietary LLMs (cheers to the authors). However, only Llama 3.1 8B is involved. I would like to see how performance changes along the spectrum of LLM scales and how large the gap is between the open-source LLMs versus the commercial black-boxed LLMs.

- In Table 2, the judgement scores for Llama 3.1 8B is 54.6, which is significantly higher than Llama 3.1 405B's 10.0 and the scores for the other LLMs. Though the authors seem to indicate (in line 353-line 354) that the model is performing randomly, therefore it achieves a random score. I am not convinced in several aspects.

    - First, how can you ensure that there is no randomness in other LLMs' answers?

    - Second, if randomly selecting a choice would lead to a much higher performance, shall we double-check the dataset to prevent such shortcuts? Do we have better ways to design the dataset to mitigate randomness or shortcuts?

    - Since there is only one 8B LLM (Llama 3.1 8B) tested, are other small (compared to larger commercial LLMs) LLMs behave similarly in terms of choosing randomly? What would be a threshold in terms of scales for LLMs to stop choosing randomly?

**Questions:**

Please see the weakness section.

---

> ### Author Response · Authors · 2025-11-21
>
> Thank you for your positive feedback on our work. We are glad that you see its value in addressing a key gap in ToM research by examining applied ToM, and we appreciate your comments on the care we took in designing concise, controlled examples that minimize confounds. We also appreciate your recognition that our findings are justified by our experiments across open-source and proprietary LLMs.
>
> **More open-source LLMs**
>
> Thank you for suggesting that we include additional open-source models. Our initial focus was to show that even state-of-the-art LLMs exhibit a substantial explicit vs. applied ToM gap, but we agree that it is valuable to examine how this pattern generalizes across the broader landscape of model scales and across open-source versus commercial models. Our original Table 2 included Llama-3.1-405B and Llama-3.1-8B as open-source models; in the revised version, we have expanded this set to include additional 1B to 14B open-source models (Llama-3.2-1B, Llama-3.2-3B, Qwen-2.5-7B, Ministral-8B, Qwen-2.5-14B, see updated Table 2 in the revision). The additional results show that this explicit vs. applied ToM gap persists across open-source models of varying scales, with smaller open-source models having even lower performance on explicit ToM as well.
>
>
> **Performance at random chance level is 50% for binary questions**
>
> We believe there may be a misunderstanding regarding our use of the phrase “around random
> performance (at 54.6%)”. From your comments, it appears this was interpreted as referring to a model’s internal randomness or stochastic sampling. However, in a binary choice classification setting, “random performance” refers to performing at chance level (50%), meaning the model exhibits no inductive bias toward the correct or incorrect option. Recognizing 50% as the random chance level baseline for binary classification (noted in version 1 Line 315) is standard in probability and evaluation methodology (generally, chance level is 100/k % where k is the number of classes).
>
> With this clarification, we address the your questions:
>
> > We are not making claims about internal randomness of LLMs. We use “random
> performance” to mean performance at random chance level and have updated our paper to make the phrasing more clear.  A model scoring far below 50% is not performing at random chance — it is systematically choosing the wrong option; a model scoring far above 50% is also not performing at random chance - it is systematically choosing the correct option; and a model scoring near 50% is performing at random chance level (such as Llama 3.1 8B at 54.6%) where it not demonstrating strong bias towards either correct or incorrect answers. This is why we interpret the 8B model as performing at random chance level, not because it contains “randomness,” but because its score is close to the chance level of 50%.
>
> > This does not imply that “randomly selecting a choice would lead to a much higher performance” nor does it suggest the presence of dataset shortcuts. A score close to 50% is random performance, and a model must be well above chance (e.g., 80–90%) to demonstrate real task capability. This follows directly from standard probability reasoning and does not reflect a dataset flaw. Additionally, SimpleToM is explicitly balanced in the correct answer distribution i.e., label distribution for correct answer was designed to be roughly equal, for mental state: 51.3% option 1 vs 48.7% option 2; behavior prediction: 51.4% option 1 vs 48.6% option2; behavior judgement: 50.1% option1 vs 49.9% option 2, removing any concern that label imbalance creates shortcuts.
>
> > Regarding small-scale models, we have expanded our experiments to include additional open-source models from 1B to 14B, which also perform below or near chance on applied ToM tasks. This shows that Llama 3.1 8B is not an isolated case and that smaller models also consistently lack reliable inductive biases needed for applied ToM reasoning.
>
> Thank you again for your thoughtful review and suggestions. We hope our additional experiments on smaller open-source models as well as clarifications on the term “random performance” address your concerns.

---

> > ### Comment · Reviewer_xnHV · 2025-11-28
> > **Response**
> >
> > I thank the authors for their comments and additional results on open-source LLMs.
> >
> > I think the primary concern for me is that in Table 2 for the main result table, the authors average the performance across mental states, behavior, and judgement. But for judgement specifically, there can be randomness associated (e.g. the model answers randomly would score ~50, while the models that are not answering randomly may score ~20). However, in the final average score, it seems that the higher the score is, the better the model is ranked, which is also the trend for the other two subcategories (mental states and behavior). This is something that bothers me on interpreting the results.
> >
> > For `randomness` specifically, what I meant is not the stochastic behavior of the model, but rather the case that for some small LLMs, there is a chance that the reasoning process for them remain incorrect (e.g. talking about something that is totally irrelevant) while still selecting the correct answer. I believe this constitutes the majority part of the Llama 3.1 8B model's answers (which the authors refer to as `randomness`). But my question is more on that for models like QWen 2.5 7B, their performance is around ~20, are there any chances that these models behave the same / similarly on certain examples? Even for larger LLMs, are these so-called `random` behaviors extinct? This was the original intent of my question.
> >
> > As of now, these two points still bother me and sorry that I am not considering raising the scores at the moment. I will raise the scores if the follow-up discussions can address these two concerns.

---

> > > ### Author Response · Authors · 2025-12-03
> > >
> > > Thank you for your follow up! We provide further clarification and details to address these two questions:
> > >
> > > **Models scoring ~20 vs ~50**
> > >
> > > Our interpretation is that a model scoring ~20% (well below the 50% chance level) is systematically failing to apply ToM inference, whereas a model scoring near ~50% shows no meaningful inductive bias toward either correct or incorrect answers.
> > >
> > > Under our statistical tests, both cases — whether a model scores ~20% or ~50% — are not significantly above the random-chance baseline of 50% (see Appendix Q for additional statistical details). This indicates that both types of models fail on the applied ToM task, differing only in whether they are systematically biased towards the wrong answer or showing no meaningful inductive bias.
> > >
> > >
> > > **Do models behave the same / similarly on certain examples?**
> > >
> > > Based on our analysis in Section 5.2 along with further details in Appendix O, model performance can vary greatly across scenarios. “Figure 14: Performance for bottom models across all scenarios” serves as a clear illustration of this and might be of interest to you. For instance, the bar graphs show that although several models like Llama-3.1.8 all perform poorly on SimpleToM, these models differ in patterns regarding their failure modes across the different scenarios. Such analysis further suggests the value of SimpleToM in providing a rigorous assessment of models across different scenarios.

---

### Official Review · Reviewer_ALBm · 2025-10-29

**Soundness:** 3
**Presentation:** 3
**Contribution:** 2
**Rating:** 4
**Confidence:** 4

**Summary:**

The paper presents SimpleToM, a two-sentence story benchmark that separates explicit ToM from applied ToM. The dataset contains 1,147 stories across 10 everyday information-asymmetry scenarios, with strict 3/3 annotator agreement filtering; a human baseline covers 50 stories / 150 questions. On SimpleToM, models achieve near-ceiling mental-state accuracy but large drops on behavior and judgment; targeted test-time guidance (mental-state reminders and ToM-specific CoT) can push strong models to ~95–97% averages.

**Strengths:**

1. The three-stage evaluation (mental state → behavior → judgment) plus first-failure analysis offers diagnostic value.
2. Two-round LLM generation, strict 3/3 agreement, and consistent retention rates across generators.
3. Clear empirical finding. Large gap between explicit and applied ToM across many models, including reasoning/inference-time models (o1/DeepSeek-R1).
4.  Mental-state reminders and CoT sharply improve applied ToM, indicating carry-over/cueing issues.

**Weaknesses:**

1. No confidence intervals, significance tests, or variance across seeds/prompts; “below random” is not visualized with chance bands.
2. Only 50 stories / 150 Qs, limiting precise human–model comparisons and per-scenario variance.
3. Binary task format may compress nuance. All tasks are binary; judgments especially could benefit from graded scales and rationales. (No direct evidence of graded/rationale analysis is provided.)
4.  Paper includes o1/DeepSeek-R1 and GPT-5 in tables, but analysis does not deeply compare inference-time design choices or situate results against recent ToM benchmarks.

**Questions:**

1. Can you add 95% CIs and paired tests/bootstraps for Tables 2/3 and error bars for Figs. 3/4 to substantiate “below random” and scenario effects?
2. Can the human baseline be expanded beyond 50 stories and reported per scenario with uncertainty?
3. Do MS and CoT gains persist on paraphrases/held-out scenarios and across reasoning-token settings for o1/DeepSeek-R1?
4. Could you incorporate graded judgments and short rationales to reduce ceiling/floor and probe value alignment? (Motivated by current binary drops.)
5. Please situate the analysis more fully within inference-based advances (o1/o3, DeepSeek-R1) and recent ToM benchmarks, clarifying why judgment lags persist despite reasoning tokens.

---

> ### Author Response · Authors · 2025-11-21
>
> We thank the reviewer for their positive feedback on the diagnostic value of our three-stage evaluation and first-failure analysis, as well as their recognition of our careful dataset construction through two-round LLM generation, strict 3/3 annotator agreement, and consistent retention across generators. We are also grateful that the reviewer found our main empirical finding clear: across many models there is a large and systematic gap between explicit ToM and applied ToM. Our further analysis in the Appendix, where question-specific mental-state reminders and ToM-specific CoT improve performance, provides further support for the reason behind this fragility we surface: models often succeed at explicit ToM but need much human guidance to carry over that information when reasoning about behavior and judgment, revealing a persistent carry-over/cueing problem.
>
> **95% CIs and paired tests/bootstraps, error bars**
>
> Thank you for your excellent suggestion – we agree that adding confidence intervals and significance tests strengthens the statistical rigor to further support our claims. In the revised version, we added 95% CIs, paired tests/bootstraps, and error bars in the per-scenario figures. These additions appear in Table 2 and are detailed extensively in Appendix Q, with further integration planned during the discussion period.
>
> **More than 50 stories / 150 Qs**
>
> We emphasize that the full 1,147-story dataset is evaluated on models, while the 50-story human subset was designed to confirm that humans do not exhibit the sharp explicit vs. applied ToM gap. Expanding human evaluation to hundreds of story scenarios is unfortunately not feasible within the discussion period timeline due to participant qualification and quality-control requirements (we would need to recruit a large pool of participant via qualification tests we had as having few participants answer many questions leads to issues like fatigue and confounding effects of repetition). Moreover, our goal is not to build a psychometric human instrument but an ML diagnostic resource, and its design intentionally prioritizes breadth of scenarios for LLM evaluation. Traditional human ToM experiments typically involve hand-designed stimuli with far fewer scenarios; one contribution of SimpleToM is precisely that LLMs enable and require large-scale evaluation across validated asymmetry types in a way human-subject studies cannot. We welcome future work that uses SimpleToM’s diverse scenarios for more extensive human-model comparisons, but they are beyond the scope of this paper.
>
> **Why binary task is more suited to reveal a jarring explicit vs applied ToM gap**
>
> We appreciate the suggestion to incorporate graded judgments or rationales. While useful for alignment studies, our aim is to reveal a striking failure mode: even on extremely clear-cut binary questions with unanimous human agreement, strong models fail to apply ToM knowledge. The binary format isolates this applied-ToM failure without confounds from rationale interpretation or scale calibration. However, we do provide ratings of harmfulness and unethicality on a scale from 1 to 5 from Claude-3.5-Sonnet and GPT-4o as described in Appendix I.3 Harmfulness and Unethicality, enabling SimpleToM to support future work on value alignment.
>
> **Inference-time reasoning models**
>
> In both our main experiments (Table 2) and preliminary tests, reasoning-enhanced models (o1/DeepSeek-R1) do not show a special advantage on applied ToM; their performance resembles non-reasoning models under generic “think step by step’’ prompting. Although inference-time interventions can sometimes improve accuracy, our analysis in Appendices F, L, M, and N shows that such interventions need to be hand-crafted, question-specific, task-specific, non-robust, and costly in terms of inference time compute. They do not provide a principled or general solution to the applied-ToM problem.
>
> **Why judgment lags persist despite reasoning tokens**
>
> Increasing reasoning tokens does not impose the multi-stage causal structure required for applied ToM (see Figure 1); while reasoning tokens help a model “think” more about the problem, but it does not mean the model would (1) think about the crucial elements for a particular task (2) apply the crucial element in their answer – this is consistent with recent findings that chain-of-thought explanations may be verbose without being algorithmically grounded (Apple ML “The Illusion of Thinking”) or unfaithful altogether (Anthropic “Reasoning models don't always say what they think”).

---

### Official Review · Reviewer_Ctoe · 2025-10-31

**Soundness:** 3
**Presentation:** 3
**Contribution:** 3
**Rating:** 6
**Confidence:** 5

**Summary:**

This paper presents SimpleToM, a benchmark that tests large language models’ Theory of Mind (ToM) abilities through 1,100 short, everyday scenarios involving natural information asymmetries. It evaluates three reasoning levels: mental state inference (explicit ToM), behavior prediction, and judgment of behavior (applied ToM). Results show that while advanced models like GPT-5 and Claude-3.5 excel at recognizing others’ beliefs, they struggle to use this understanding for predicting or judging actions. This highlights a key gap between knowing and applying social reasoning, suggesting that future AI work must focus on bridging this explicit–applied ToM divide.

**Strengths:**

- The paper makes a valuable distinction between explicit and applied Theory of Mind (ToM), highlighting an overlooked but critical aspect of social reasoning in LLMs.
- SimpleToM covers diverse, realistic scenarios with naturally occurring information asymmetries, improving ecological validity over previous ToM datasets.
- High data quality: The combination of LLM-assisted story generation and rigorous human annotation yields concise, well-controlled, and reliable test cases.
- Results across 16 frontier models provide a broad and insightful comparison of current ToM capabilities and their limitations.

**Weaknesses:**

- The benchmark’s tightly templated two-sentence stories and binary question formats (yes/no, two-option, or “reasonable/unreasonable”) may oversimplify the complexity of social reasoning. This constrained setup may produce below-random accuracies in judgment tasks for strong models, likely reflecting design artifacts rather than genuine reasoning deficits.
- The “judgment” task assumes a single normative ground truth derived from crowdworker consensus. Because moral or social appropriateness varies culturally and contextually, this approach embeds hidden biases that could distort conclusions about model morality or social reasoning.
- Many stories rely on repeated structural motifs (e.g., containers, hidden information) that may conflate Theory of Mind reasoning with world-knowledge heuristics, making it unclear whether models are failing at perspective-taking or at commonsense inference.
- The study’s interventions (prompt reminders, system messages, chain-of-thought prompts) are limited and not systematically ablated, so the paper’s claim that the explicit–applied ToM gap is robust across prompting strategies remains only partially supported

**Questions:**

- Have you tried the interventions on reasoning models?
- The paper analyzes “first failure” along the MS → behavior → judgment chain. Do you observe non-monotonic cases (e.g., behavior correct when MS is wrong), and if so, how frequent are such patterns per scenario?

---

> ### Author Response · Authors · 2025-11-21
>
> We thank the reviewer for their positive feedback and for recognizing SimpleToM’s contributions: distinguishing explicit vs. applied Theory of Mind (ToM), covering diverse everyday scenarios with naturally occurring information asymmetries, ensuring high data quality, and evaluating a broad range of frontier models. We especially appreciate the reviewer’s recognition that separating explicit from applied ToM exposes a fundamental but previously overlooked limitation of current LLMs.
>
> **Two-sentence stories and binary question formats**
>
> The minimalist design is intentional: it isolates whether models can apply mental-state information at all. Humans agree unanimously on these items (3/3 annotators), yet strong models still fail at the applied-ToM stages — direct evidence of a core applied-ToM inability. Richer or longer stories would introduce additional factors (memory, distractors, narrative complexity) that obscure the specific failure mode our benchmark targets. Guided by psychological criteria for valid ToM testing (Quesque & Rossetti 2020; Chen et al. 2024), we deliberately use a simple, controlled setup to avoid such confounders.
>
> Below-random judgment performance is not a design artifact. As shown in Appendix M and Fig. 11, even when models are reminded of their own mental-state answer, they often fail to apply it unless given custom, ToM-specific CoT prompts (generic CoT is insufficient). Further, we do not observe this jarring gap in human annotators using the exact same phrasing, supporting the interpretation that these failures reflect genuine applied-ToM deficits in LLMs.
>
> **Single normative ground truth**
>
> The judgment task does not aim to measure morality or cultural alignment. It assesses whether a model can evaluate the internal coherence of a character’s behavior considering their beliefs. All our stories present situations that are clear-cut with unanimous human agreement, minimizing normative variability. While this paper focuses on the explicit applied vs ToM gap, we also provide harmfulness/unethicality ratings on a scale from 1 to 5 (Appendix I.3) to support future value-alignment studies.
>
> **Commonsense vs. ToM**
>
> A key design feature of SimpleToM is that scenarios require models to perform commonsense inference rather than rely on explicit mentalizing words to determine what each character can perceive or believe. This mirrors how humans naturally make ToM inferences in real life (people do not explicitly say that “the Pringles can is opaque” before doing so; we infer such facts automatically), making the task more natural and cognitively realistic.
>
> Our three-stage evaluation (MS → behavior → judgment) disentangles at which stage models fail at: strong models almost always answer the mental-state question correctly showing they can perform the required commonsense inference and mental state attribution (models like GPT-4o, Llama-3.1-405B, Claude-3-Opus, GPT-4, GPT-5, Claude-3.5-Sonnet, o1-preview, o1, DeepSeek-R1 all achieved accuracies of more than 95%) yet their performance drop sharply on behavior and judgment, where applying that inferred mental state is necessary. This dissociation demonstrates that the observed gaps reflect applied ToM failures, not world-knowledge confusion.
>
>
> **Prompting strategies**
>
> Our main contribution is to expose the explicit vs applied ToM gap without specialized prompting. Appendices F, L, M, and N show that interventions require hand-crafted, question-specific, ToM-specific prompts, which are non-robust and costly in terms of inference time compute. They do not constitute a principled solution.
>
> **Reasoning models**
>
> In our early experiments, we tested o1 and o1-preview with MS reminders and CoT. Their applied-ToM performance remains comparable to non-reasoning models with generic “think step by step” prompts as in our main results. We do not introduce this distinction between reasoning and non-reasoning models beyond main results in Table 2 – while reasoning tokens help a model “think” more about the problem, but it does not mean the model would (1) think about the crucial elements for a particular task (2) apply the crucial element in their answer – this is consistent with recent findings that chain-of-thought explanations may be verbose without being algorithmically grounded (https://machinelearning.apple.com/research/illusion-of-thinking) or unfaithful altogether (https://www.anthropic.com/research/reasoning-models-dont-say-think).
>
>
> **Non-monotonic cases**
>
> Non-monotonic patterns—e.g., later-stage correctness despite earlier-stage errors—are rare across strong models:
>
> Llama-3.1-405B: 14 cases (1.2%)
>
> Claude-3.5-Sonnet: 39 cases (3.4%)
>
> GPT-4o: 58 cases (5.1%)
>
> The slightly common pattern is MS correct → BP incorrect → JU correct, while BP correct when MS is wrong is extremely rare. These numbers reinforce that models generally do infer mental states but struggle to apply them consistently.

---

### Author Response · Authors · 2025-12-03
**General Response**

We thank the reviewers for their helpful suggestions, which strengthened the paper in several ways. In particular, we have:

- Added more open-source models across different sizes (7B–13B), confirming that the explicit vs. applied ToM gap persists across scale and architecture. (Table 2; Results section)
- Included 95% CIs, paired tests, bootstraps, and error bars, strengthening the statistical support for below-chance effects and scenario-level differences. (Table 2; main text; Appendix Q)
- Expanded our analysis of inference-time reasoning, adding illustration (Figure 5), elaboration, and citation of related work to show why such methods are unlikely to help with applied ToM. Our results (Appendices F, L, M, N) further demonstrate that improvements require hand-crafted, ToM-specific guidance that does not generalize, highlighting that the explicit vs. applied ToM gap reflects a fundamental limitation of current models and the fragility of current SOTA LLMs.
- Provided analysis of non-monotonic cases, showing they are rare (1–5%), further confirming failures stem from the explicit vs applied ToM gap.


We also appreciate the reviewers for their thoughtful engagement with our work and are grateful that they recognized the key strengths of our SimpleToM dataset:

- Presenting a novel diagnostic benchmark that **isolates explicit vs applied ToM**, requiring models to not only infer mental states (explicit ToM) but also apply them to behavior and judgment (applied ToM).
- Covering a **broad range of real-world information asymmetries** — including deception, secrecy, manipulation, hidden flaws — in **diverse everyday scenarios**, far beyond prior single-template benchmarks.
- Using scenarios that require commonsense inference rather than explicit mentalizing cues, **mirroring how humans naturally infer what characters can perceive or believe** (e.g., no one explicitly states “the Pringles can is opaque”).
- **Drawing directly from psychology** in (1) following psychological criteria for valid ToM testing, using a simple, controlled setup that avoids confounders such as memory load, distractors, and narrative complexity, (2) incorporating well-established distinctions in ToM stages/components (three-stage MS→behavior→judgment) that prior LLM benchmarks overlook.
- **Ensuring realism and logical consistency** through strict 3/3 human filtering, removing flaws that appear in existing datasets.

We are also pleased that the reviewers highlighted the significance of our experimental findings:

- Revealing a striking, previously unseen gap: frontier models achieve ≥95% mental-state accuracy but fail on applied ToM even in two-sentence, unambiguous scenarios — unlike humans on the same stimuli.
- Providing strong diagnostic value, exposing (1) the explicit vs applied ToM gap, (2) large variation across scenarios, and (3) the limits of prompting-based fixes for this fundamental applied ToM gap — limitations that prior datasets could not surface.

These elements together constitute the key novelty of our work, and the thoroughness of our write-up lay a rigorous foundation for driving future advances toward closing the fundamental explicit vs applied ToM gap in next-generation models.

---

### Meta-Review · Area_Chair_mPRo · 2026-01-19

**Summary:**

This paper proposes SimpleTOM, a benchmark for evaluating LLMs' theory-of-mind abilities. The paper shows that while SOTA models are good at predicting the mental state, they are not good at applying that for behaviour prediction and judgment tasks.

Reviewers had several concerns, including the interpretation of results and the novelty of the benchmark. The authors have addressed all the concerns in their rebuttal.

Overall, this paper highlights one of the major limitations of LLMs with a simple and clear benchmark with as few confounding factors as possible. So I recommend accepting this paper.

**Reviewer Concerns:**

1. There was a concern about novelty with respect to BigToM benchmark, but authors have clearly explained the differences and I agree with their response.
2. Reviewers asked for confidence interval and other improvements to results which are all provided during the rebuttal.
3. All the other concerns were promptly addressed by the authors.

**Reviewer Scores:**

Reviewer Ctoe would stay at 6.
Reviewer ALBm would have updated their score from 4 to 6.
Reviewer xnHV would have updated their score from 4 to 6.
Reviewer 6erB might have kept their score at 4.

---

### Decision · Program_Chairs · 2026-01-26

Accept (Poster)